# Development of the terminal air spaces in the gray short-tailed opossum (*Monodelphis domestica*)– 3D reconstruction by microcomputed tomography

**Kirsten Ferner** *

Department Evolutionary Morphology, Leibniz-Institut für Evolutions- und Biodiversitätsforschung, Museum für Naturkunde, Berlin, Germany

* kirsten.ferner@mfn.berlin

## Abstract

Marsupials are born with structurally immature lungs when compared to eutherian mammals. The gray short-tailed opossum (*Monodelphis domestica*) is born at the late canalicular stage of lung development. Despite the high degree of immaturity, the lung is functioning as respiratory organ, however supported by the skin for gas exchange during the first postnatal days. Consequently, the majority of lung development takes place in ventilated functioning state during the postnatal period. Microcomputed tomography (µCT) was used to three-dimensionally reconstruct the terminal air spaces in order to reveal the timeline of lung morphogenesis. In addition, lung and air space volume as well as surface area were determined to assess the functional relevance of the structural changes in the developing lung. The development of the terminal air spaces was examined in 35 animals from embryonic day 13, during the postnatal period (neonate to 57 days) and in adults. At birth, the lung of *Monodelphis domestica* consists of few large terminal air spaces, which are poorly subdivided and open directly from short lobar bronchioles. During the first postnatal week the number of smaller terminal air spaces increases and numerous septal ridges indicate a process of subdivision, attaining the saccular stage by 7 postnatal days. The 3D reconstructions of the terminal air spaces demonstrated massive increases in air sac number and architectural complexity during the postnatal period. Between 28 and 35 postnatal days alveolarization started. Respiratory bronchioles, alveolar ducts and a typical acinus developed. The volume of the air spaces and the surface area for gas exchange increased markedly with alveolarization. The structural transformation from large terminal sacs to the final alveolar lung in the gray short-tailed opossum follows similar patterns as described in other marsupial and placental mammals. The processes involved in sacculation and alveolarization during lung development seem to be highly conservative within mammalian evolution.

**Data Availability Statement:** The data that support the findings of this study, original images of the figures and further images and videos of 3D reconstructions of the terminal air spaces are made

publicly available with figshare (data: https://doi.org/10.6084/m9.figshare.24764187; original images: https://doi.org/10.6084/m9.figshare.24763497; 3D-images: https://doi.org/10.6084/m9.figshare.24771213 and 3D-videos: https://doi.org/10.6084/m9.figshare.24764397).

**Funding:** The author received funding from the German Research Foundation (DFG) with the module "temporary position for the principal investigator" (Grant No. FE1878/2-1). The funders had no role in study design, data collection and analysis, decision to publish, or preparation of the manuscript. Open access funding was enabled by the Leibniz Association's Open Access Publishing Fund.

**Competing interests:** The authors have declared that no competing interests exist.

## Introduction

Marsupials have a unique reproductive strategy compared to placental mammals. The early stage of development at birth and the subsequent slow postnatal development attached to the maternal teat is a characteristic feature of marsupials that distinguishes them from other mammals [1]. The marsupial gray short-tailed opossum (*Monodelphis domestica*) is born approximately 13.5 days after conception in a highly immature, nearly embryonic condition. The neonate is extremely small (130 mg) and exhibits a minimum anatomical development possible for a newborn mammal at birth. Most of the organ systems are immature, e.g., a functioning mesonephros, liver with simple sinusoid system, cartilaginous skeleton [2]. However, in adaptation to the reproductive strategy the neonate appears to be well developed in certain aspects. An advanced olfactory system, well pronounced forelimbs and a muscular brachial plexus allow the neonate to crawl from the vagina to the mammary patch, attach itself to a maternal teat and start to suckle immediately.

Compared to eutherian neonates, marsupials are generally born with structurally immature lungs at the canalicular or saccular stage [3–18]. The gray short-tailed opossum is born with lungs at the canalicular stage of lung development [17]. Consequently, the majority of lung development occurs postnatally in air attached to the maternal teat.

While the lung in marsupials appears structurally immature, it shows qualitative characteristics of a mature gas-exchanging organ, e.g., surfactant [13,19,20], a thin blood-gas barrier [8,15], neuronal-muscular reflex control of breathing [21]. Thus, from the viewpoint of passive mechanics there might be no major constraints to inspiration [13]. However, poor muscle coordination and chest-wall distortion cause severe constraints to pulmonary ventilation [22]. These neural and mechanical constraints at birth necessitate recruitment of an alternative organ system such as the skin for gas exchange [10,21–24]. Cutaneous respiration is enabled by a subcutaneous capillary network with short air-blood diffusion distances, a large surface area to volume ratio, low metabolic rate and the presence of cardiac shunts in these immature newborns [16,21,25–27]. Even though supported by cutaneous respiration, most marsupials have functioning lungs at birth and rely on them as major gas exchanging organ.

The mammalian lung development was mainly studied in eutherian species, e.g., in mice and rats, and can be categorized into five morphological stages (embryonic, pseudoglandular, canalicular, saccular and alveolar) based on characteristic morphology [28–32].

Lung development starts in the embryonic stage (prenatal days E11–E13 in rats) with the formation of the two lung buds. At the terminal ends of the buds, a repetitive process starts where elongation of the future airways alternates with branching. The major airways and the pleura are formed. In the pseudoglandular stage (E13–E18.5 in rats) the preacinar branching pattern of airways and blood vessels is established [33,34]. The following canalicular stage (E18.5–E20 in rats) is characterized by branching of the terminal bronchi, terminating in small canaliculi and differentiation of type I and type II alveolar epithel cells (AECs). Towards the end of this period, the terminal or acinar tubes narrow and give rise to small saccules. Epithelial differentiation and angiogenetic activation of the capillaries lead to the first functional air-blood barriers in the lung [21,35,36]. The development of the lung proceeds with the saccular stage (E20 to postnatal day 4 in rats), which is characterized by saccule expansion, tissue proliferation, septal development and remodeling. During the alveolar stage respiratory airways and acini develop [37]. The gas-exchange area is further enlarged by lifting off new septa from the existing gas-exchange surface and subdivision of the terminal air spaces [32,38]. During microvascular maturation, the double-layered capillary network of the alveolar septa is reduced to a single-layered one to increase the efficiency of the lung [39–41].

Alveolarization can be divided into two distinct phases and continues in the postnatal period [42,43]. During classical alveolarization (postnatal day 4–21 in rats), new septa are formed from preexisting immature septa which contain a double-layered capillary network. During continued alveolarization (P14 to approximately P60 in rats), new septa are formed from preexisting mature single-layered capillary septa.

In eutherians, the majority of lung development occurs throughout intrauterine life. The lungs of most newborn eutherians are at the alveolar stage and the key changes that occur during early postnatal life include the increase of alveolar number and maturation of microvasculature [32,41,44]. Only very altricial eutherian neonates, such as mice, rats and shrews are born at the saccular stage, but reach the alveolar stage during the first postnatal days [15,40,45].

In contrast to eutherian neonates, marsupials go through most of their lung development in the postnatal period. The developmental degree of the lung in newborn marsupials corresponds to the Carnegie stage 16–17 in the human fetus or E13–E14 in the fetal rat [46].

In recent years the establishment of μCT techniques in combination with 3D reconstruction allowed to examine the three-dimensional structure of the lung and provided insight into the alveolarization of mouse and rat lungs [30,36,42,47]. So far, only one study examined the developing lung of two marsupial species by phase contrast imaging methods [16]. The 3D reconstruction of the lungs revealed that only two lung sacs were present in the newborn fat-tailed dunnart, whereas the lungs of the tammar wallaby had numerous large terminal saccules. Both species undergo marked increases in architectural complexity during the postnatal period [16].

Studies on comparative lung development in various mammalian species let assume that mammalian lung development is highly conservative and follows similar developmental pathways in all mammalian species, including marsupials and monotremes [2,15,48,49]. The stage of lung development when mammals are born is quite variable, but the sequence of developmental steps resulting in final lung maturation are not. *Monodelphis domestica* resembles both the supposed marsupial and mammalian ancestor [2,44,50]. Its ancestral condition and the finding that, in contrast to eutherian mammals, most of the lung development occurs postnatally in a ventilated functioning state offers a unique opportunity for a better understanding of the development of the mammalian lung.

As a first step, the development of the bronchial tree in *Monodelphis domestica* was investigated [51]. The present study was targeted at the development of the terminal air spaces of the lung in the gray short-tailed opossum during the postnatal period using microcomputed tomography (μCT). In addition, we aimed to obtain functional volumes of the air spaces and surface areas of the lung using three-dimensional (3D) reconstructions of computed tomography (CT) data.

## Material and methods

### Animal collection

Gray short-tailed opossums from a laboratory colony established at the Museum für Naturkunde Berlin (Berlin, Germany) were controlled mated for this study. The females were checked for offspring when approaching full-term (13–14 days). Young ranging from birth to 57 days post natum (dpn) and adults (primi- or multiparous females one year old) were collected, weighed and euthanized by anaesthetic overdose with isoflurane under animal ethics permit approved by the Animal Experimentation Ethics Committee (registration number: T0202/18). To assess possible changes around parturition, one female was euthanized shortly before term by day 13 of gestation and the embryos were dissected and fixed by Karnovsky fixative [52]. A total of 35 animals between 13 days post coitum (dpc) and adult were studied. Additional eight animals (Neonate, 5, 7,14, 21, 28, 56 dpn and adult) were investigated by scanning electron microscopy (SEM). All available details of the specimens are given in Table 1.

## Sample preparation

Early developmental stages, ranging from neonate (defined as the first 24 h at the day of birth) to 28 dpn were decapitated, to allow for lung fixation via the trachea. The whole body of the animals was immediately immersed in Karnovsky fixative (2 g paraformaldehyde, 25 ml distilled water, 10 ml 25% glutaraldehyde, 15 ml 0.2 M phosphate buffer) or Bouin's solution (picric acid, formalin, 100% acetic acid, 15: 5: 1; [52]). The fixation time in Bouin was usually one to two days. Afterwards the specimens were rinsed in 70% ethanol. Specimens fixed with Karnovskys fixative stayed in the fixative until scanning (between some weeks and months). In late developmental stages, from 35 dpn to adults, the lungs were fixed by instillation via the trachea and finally dissected. Karnovsky fixative was inserted in the trachea via a cannula with polyethylene catheter tubing at a pressure head of 20 cm, until fixative was emerging from nostrils and mouth.

Some animals were processed for transmission and scanning electron microscopy (TEM and SEM). The TEM samples were used for further ultrastructural analysis, which is not subject to this study. From neonate to 11 dpn the upper part of the trunk was cut in two halves and in older stages (14 dpn–adult) the lungs were dissected.

The specimens for electron microscopy were fixed in 2.5% glutaraldehyde buffered in 0.2 M cacodylate (pH 7.4) for 2 hours, rinsed with 0.1 M cacodylate buffer and either postfixed in 1% osmium tetroxide and embedded in epoxy resin (Araldite) for TEM or dried in alcohol (30–100%), 'critical-point-dried', mounted, sputter-coated with gold-palladium for SEM. The samples were viewed and photographed in a scanning electron microscope (LEO 1450 VP, Carl Zeiss NT GmbH) to see ultrastructural details of the 3D architecture of the lung. Details of the specimens investigated by SEM are given in Table 2.

## Preparation for μCT imaging

Comparative, functional, and developmental studies of animal morphology require accurate visualization of three-dimensional structures, but few widely applicable methods exist for non-destructive whole-volume imaging of animal tissues. μCT imaging in comparative morphology has been used in paleontology, where mineralized tissue, e.g., bones, were scanned. However, μCT-imaging of soft-tissue structures has been limited by the low intrinsic x-ray contrast of non-mineralized tissues. With sufficient contrast imparted to soft tissues, organs, such as lung, liver, kidney, heart, intestine, skin and brain, can be made visible with μCT- techniques. With very simple contrast staining μCT imaging produces quantitative, high-resolution, high-contrast volume images of lung tissue. This is possible without destroying the specimens and with possibilities of combining with other preparation and imaging methods (histology or TEM).

In [53] several simple and versatile staining methods for μCT-imaging of animal soft tissues are summarized, along with advice on tissue fixation and sample preparation. Based on this information, different staining protocols using inorganic iodine and phosphotungstic acid (PTA), were developed, tested and used to produce high-contrast x-ray images of the lung at different age stages (Table 1).

Staining with PTA was either performed in ethanol with 1% PTA for 21 to 42 days (full body specimens of 14 to 28 dpn) or in an aqueous solution (Karnovsky fixative) starting with 0.5% for 7 to 20 days and increased afterwards to 1% resulting in a staining period up to 30 days [53]. Separated lungs of older stages and adult specimens were stained in 1% Iodine [54] to test staining differences and shrinking effects, which could not be detected. The differences in staining periods and staining concentration depended on the respective specimen size and preparation.

Torsos and lungs were scanned in distilled water using a small container. The specimens were fixed in the container with cotton balls to avoid moving around during the scan.

**Table 1. Gray short-tailed opossum (*Monodelphis domestica*) specimens examined in this study. Body weights, air space diameter, septum thickness, volumes of the lung and terminal air spaces and surface area are presented.**

| Age (days) | No. | Medium | Staining | BW (g) | Air space diameter (μm) | Septum thickness (μm) | Lung volume, $V_L$ (mm³) | Terminal air space volume, $S_A$ (mm³) | Surface area, $S_A$ (mm²) |
|---|---|---|---|---|---|---|---|---|---|
| **13 dpc** | 2095d* | A. | - | - | - | - | 0.63 | - | - |
| | 2095e | K. | PTA | - | - | - | 0.48 | - | - |
| | 2095f* | A. | - | - | - | - | 0.36 | - | - |
| | 2095g | K. | PTA | - | - | - | 0.66 | - | - |
| | **Mean (SD)** | | | | - | - | - | **0.53 (±0.14)** | - | - |
| **Neonate** | 2350_1 | K. | PTA | 0.13 | 294 (±70) | 42 (±11) | 1.73 | 0.57 | 42.68 |
| | 2350_3 | K. | PTA | 0.13 | 351 (±97) | 40 (±11) | 1.98 | 0.91 | 40.14 |
| | 2350_7 | K. | PTA | 0.13 | 403 (±87) | 41 (±13) | 2.74 | 1.39 | 46.95 |
| | **Mean (SD)** | | | **0.13 (±0.00)** | **349 (±55)** | **41 (±1)** | **2.15 (±0.53)** | **0.96 (±0.41)** | **43.26 (±3.44)** |
| **4 dpn** | 2257_4* | A. | - | 0.21 | 259 (±51) | 33 (±10) | 4.35 | 1.74 | 100.99 |
| | 2257_6 | K. | PTA | 0.21 | 258 (±76) | 34 (±9) | 4.19 | 1.64 | 87.85 |
| | 2257_3* | A. | - | 0.21 | 257 (±77) | 34 (±12) | 4.83 | 2.00 | 106.70 |
| | **Mean (SD)** | | | **0.21 (±0.00)** | **258 (±1)** | **34 (±0)** | **4.46 (±0.33)** | **1.79 (±0.19)** | **98.51 (±9.67)** |
| **7 dpn** | 2383_2 | K. | PTA | 0.28 | 196 (±57) | 25 (±11) | 5.63 | 2.29 | 153.16 |
| | 2383_4* | A. | - | 0.27 | 187 (±80) | 22 (±8) | 5.56 | 2.23 | 182.87 |
| | **Mean (SD)** | | | **0.28 (±0.01)** | **192 (±6)** | **24 (±2)** | **5.60 (±0.05)** | **2.26 (±0.04)** | **168.02 (±21.01)** |
| **11 dpn** | 1993_2 | E. | Iodid | 0.69 | 148 (±38) | 25 (±7) | 20.58 | 4.68 | 273.14 |
| | 2419_3* | A. | - | 0.45 | 140 (±46) | 26 (±10) | 14.84 | 5.92 | 227.73 |
| | 1993_3* | A. | - | 0.73 | 135 (±40) | 23 (±10) | 21.42 | 4.77 | 317.88 |
| | **Mean (SD)** | | | **0.62 (±0.15)** | **141 (±7)** | **25 (±2)** | **18.95 (±3.58)** | **5.12 (±0.69)** | **272.92 (±45.08)** |
| **14 dpn** | 1994_8 | E. | PTA | 0.99 | 132 (±30) | 26 (±6) | 27.78 | 9.98 | 409.45 |
| | 1994_9 | E. | PTA | 0.98 | 137 (±28) | 27 (±5) | 24.87 | 12.11 | 699.04 |
| | 1994_10 | E. | PTA | 1.03 | 110 (±28) | 24 (±7) | 24.64 | 7.72 | 368.44 |
| | **Mean (SD)** | | | **1.00 (±0.03)** | **126 (±14)** | **26 (±2)** | **25.76 (±1.75)** | **9.94 (±2.19)** | **492.31 (±180.20)** |
| **21 dpn** | 2040 | E. | PTA | 2.43 | 106 (±23) | 23 (±3) | 71.14 | 20.88 | 877.41 |
| | 2037* | A. | - | 2.34 | 116 (±29) | 16 (±4) | 58.84 | 23.84 | 1220.42 |
| | 2036* | A. | - | 2.26 | 101 (±26) | 16 (±4) | 51.42 | 23.68 | 1755.55 |
| | **Mean (SD)** | | | **2.34 (±0.09)** | **108 (±8)** | **18 (±4)** | **60.47 (±9.96)** | **22.80 (±1.66)** | **1284.46 (±442.56)** |
| **28 dpn** | 2059 | E. | PTA | 4.22 | 80 (±14) | 19 (±5) | 208.62 | 68.79 | 3570.22 |
| | 2060 | E. | Iodid | 4.16 | 98 (±18) | 20 (±5) | 189.75 | 76.69 | 3676.58 |
| | **Mean (SD)** | | | **4.19 (±0.04)** | **89 (±13)** | **20 (±0)** | **199.19 (±13.34)** | **72.74 (±5.59)** | **3623.40 (±75.21)** |
| **35 dpn** | 2065 | E. | Iodid | 7.25 | 85 (±18) | 17 (±5) | 388.59 | 143.07 | 9606.84 |
| | 2405 | K. | PTA | 6.08 | 61 (±15) | 10 (±3) | 344.00 | 144.51 | 7616.83 |
| | 2194 | K. | PTA | 7.68 | 64 (±16) | 20 (±4) | 493.68 | 184.93 | 9982.44 |
| | **Mean (SD)** | | | **7.00 (±0.83)** | **70 (±13)** | **16 (±5)** | **408.76 (±76.85)** | **157.50 (±23.76)** | **9068.70 (±1271.31)** |
| **49 dpn** | 2049 | E. | Iodid | 13.64 | 58 (±10) | 16 (±4) | 504.03 | 201.78 | 13703.22 |
| | 2402 | K. | PTA | 11.59 | 50 (±12) | 9 (±2) | 452.65 | 237.29 | 15529.72 |
| | 2403 | K. | PTA | 11.09 | 47 (±11) | 10 (±2) | 408.96 | 236.15 | 16672.78 |

*(Continued)*

**Table 1.** (Continued)

| Age (days) | No. | Medium | Staining | BW (g) | Air space diameter (μm) | Septum thickness (μm) | Lung volume, $V_L$ (mm³) | Terminal air space volume, $S_A$ (mm³) | Surface area, $S_A$ (mm²) |
|---|---|---|---|---|---|---|---|---|---|
| | **Mean (SD)** | | | **12.11 (±1.35)** | **52 (±6)** | **12 (±4)** | **455.22 (±47.59)** | **225.07 (±20.18)** | **15301.91 (±1497.83)** |
| **57 dpn** | 2179 | E. | Iodid | 31.58 | 63 (±11) | 9 (±2) | 1176.68 | 372.105 | 18896.72 |
| | 2413 | K. | PTA | 14.81 | 44 (±9) | 9 (±2) | 663.60 | 353.06 | 22073.97 |
| | 2416 | K. | PTA | 18.53 | 47 (±9) | 12 (±2) | 790.89 | 352.45 | 24499.42 |
| | **Mean (SD)** | | | **21.64 (±8.81)** | **51 (±10)** | **10 (±2)** | **877.06 (±267.17)** | **359.21 (±11.18)** | **21823.37 (±2809.74)** |
| **Adult** | 2095 | K. | PTA | 87.22 | 61 (±15) | 11 (±3) | 2948.94 | 1396.63 | 43744.40 |
| | 2117 | K. | PTA | 69.47 | 81 (±16) | 10 (±3) | 2307.41 | 914.69 | 31591.77 |
| | 2419 | K. | PTA | 66.46 | 94 (±20) | 10 (±3) | 2631.63 | 1387.20 | 40383.30 |
| | **Mean (SD)** | | | **74.38 (±11.22)** | **79 (±17)** | **10 (±1)** | **2629.33 (±320.77)** | **1232.84 (±275.57)** | **38573.16 (±6275.27)** |

A., Araldite; BW, body weight; dpc, days post coitum; dpn, days post natum; E., 70% Ethanol (after fixation with Bouin); K., Karnovsky fixative; PTA, phosphotungstic acid; * Specimen processed for Transmission electron microscopy.

The specimens processed for TEM were scanned without further processing. The osmium fixation during the TEM processing led to intensive staining of the tissue and resulted in well contrasted scans. The Araldite blocks were fixed on a glass plate with glue to avoid moving during the scans. Three blocks at a time could be mounted and scanned consecutively with the best resolution for each sample respectively. The scanning results of the Araldite blocks depended on proper fixation of the TEM specimens. The μCT scans of the Araldite blocks offer a great possibility to check the quality of TEM specimens before performing expensive and time-consuming ultra sections.

## μCT imaging

The prepared specimens were subjected to μCT analysis at the Museum für Naturkunde Berlin (lab reference ID SCR_022585) using a Phoenix nanotom X-ray machine (Waygate Technologies, Baker Hughes, Wunstorf, Germany; equipment reference ID SCR_022582). It was running at 70–110 kV and 75–240 μA, generating 1440–2000 projections (Average 3–6) with

**Table 2.** Gray short-tailed opossum (*Monodelphis domestica*) specimens examined by scanning electron microscopy.

| SEM specimen | No. | Fixation | BW (g) | Lung sections |
|---|---|---|---|---|
| Neonate | 138 | Immersion Ga./Cb. | 0.12 | left and right lung |
| 5 dpn | 269 | Immersion Ga./Cb. | 0.21 | left and right lung |
| 7 dpn | 1102 | Installation Ga./Cb. | 0.35 | left and right lung |
| 14 dpn | 1104 | Installation Ga./Cb. | 0.90 | left and right lung |
| 21 dpn | 1080 | Installation Ga./Cb. | 2.58 | left and right lung |
| 28 dpn | 1058 | Installation Ga./Cb. | 4.36 | left and right lung |
| 56 dpn | 1061 | Installation Ga./Cb. | 23.20 | right lung |
| Adult | 1030 | Installation Ga./Cb. | 65.24 | left and right lung |

Cb., Cacodylate buffer; BW, body weight; dpn, days post natum; Ga., Glutaraldehyde.

750–1000 ms per scan. For bigger specimens a YXLON FF85 (equipment reference ID SCR_020917) was used. It operated with a transmission beam at 90–110 kV and 100–150 µA, generating 2000 projections (Average 3) with 250–500 ms. The different kV, µA and projection-settings depended on the respective machine and specimen size, which is also responsible for the range of the effective voxel size between 1.5–20.1 µm. The cone beam reconstruction was performed using the datos|x 2 reconstruction software (Waygate Technologies, Baker Hughes, Wunstorf, Germany; datos|x 2.2).

## Segmentation, visualisation and data analysis for 3D reconstruction

Non-destructive µCT-imaging, in particular of entire animals, offers various possibilities for different research approaches. Surface scans give an impression of the external anatomy of the examined animals (Fig 1A–1C) or the anatomical position of 3D reconstructed organs can be assessed (Fig 1D).

The 3D volume processing was carried out with the software Volume Graphics Studio Max Version 3.5 (Volume Graphics GmbH, Heidelberg, Germany). µCT data were analyzed as serial two-dimensional (2D) and reconstructed to three-dimensional (3D) images (Fig 2).

The segmentation was carried out on 16-bit images to reconstruct the entire air spaces of the lung. A region grower tool was used, that marks all areas of the same density-value connected to each other to create a region of interest (ROI). The tissue density is mapped to gray values, so that tissues of the same density appear in the same gray scale value. A tolerance of 1000–1200 gray scale values around the first selected gray value of the ROI (center of the trachea) was given. Starting from the centerline of the trachea, the region grower tool was extended to the tracheal wall. From there the ROI was extended by scrolling through the image stack and applying region growing to the airway walls and subsequently to the terminal air space walls. It was visually ensured that only air spaces were included. In that way pulmonary blood vessels and other air-filled areas in or between the lung segments were excluded from the segmentation. Calculations of volume and surface area are built-in functions of Volume Graphics Studio Max. With segmentation a ROI will be created, which has a certain volume and surface area. The first ROI "bronchial tree" contained the entire bronchial tree of the lung, beginning from the trachea and extrapulmonary main bronchi to the terminal bronchioles [51]. The surface area and volume of the ROI "bronchial tree" was calculated by the program. In a next step the ROI of the bronchial tree was copied and then extended to include the terminal air spaces. The resulting ROI "entire air spaces" included all conducting and terminal air spaces of the lung. Volume and surface area were determined for the ROI "entire air spaces". By subtracting the surface area and volume of the ROI "bronchial tree" from the ROI "entire air spaces", the surface area ($S_A$) and volume of the terminal air spaces ($V_A$) was determined. Not all surfaces, especially in the later postnatal stages, might be reproduced perfectly. This could have led to an artificial roughness in segmentations, which might also influence the calculations of surface areas and volume of air spaces. In this case, the surface area might be overestimated and the volume of terminal air spaces would be lower than in reality. However, comparable literature data reported for other marsupial species show the robustness of the obtained values [9,12,14,16,18].

For estimating the lung volume ($V_L$) the outline of the lung, excluding the extrapulmonary main bronchi, was segmented and $V_L$ was calculated from the ROI "entire lung". The values for $V_L$, $V_A$ and $S_A$ are presented as mean and standard deviation in Table 1. For graphical representation, lung volume, air space volume and surface area in relation to body mass are shown as bilogarithmic plots (Fig 3). For comparison the data of *Monodelphis domestica* are plotted together with literature data from other marsupial and eutherian species. The graphs are based on individual animal data and the regression lines are provided.

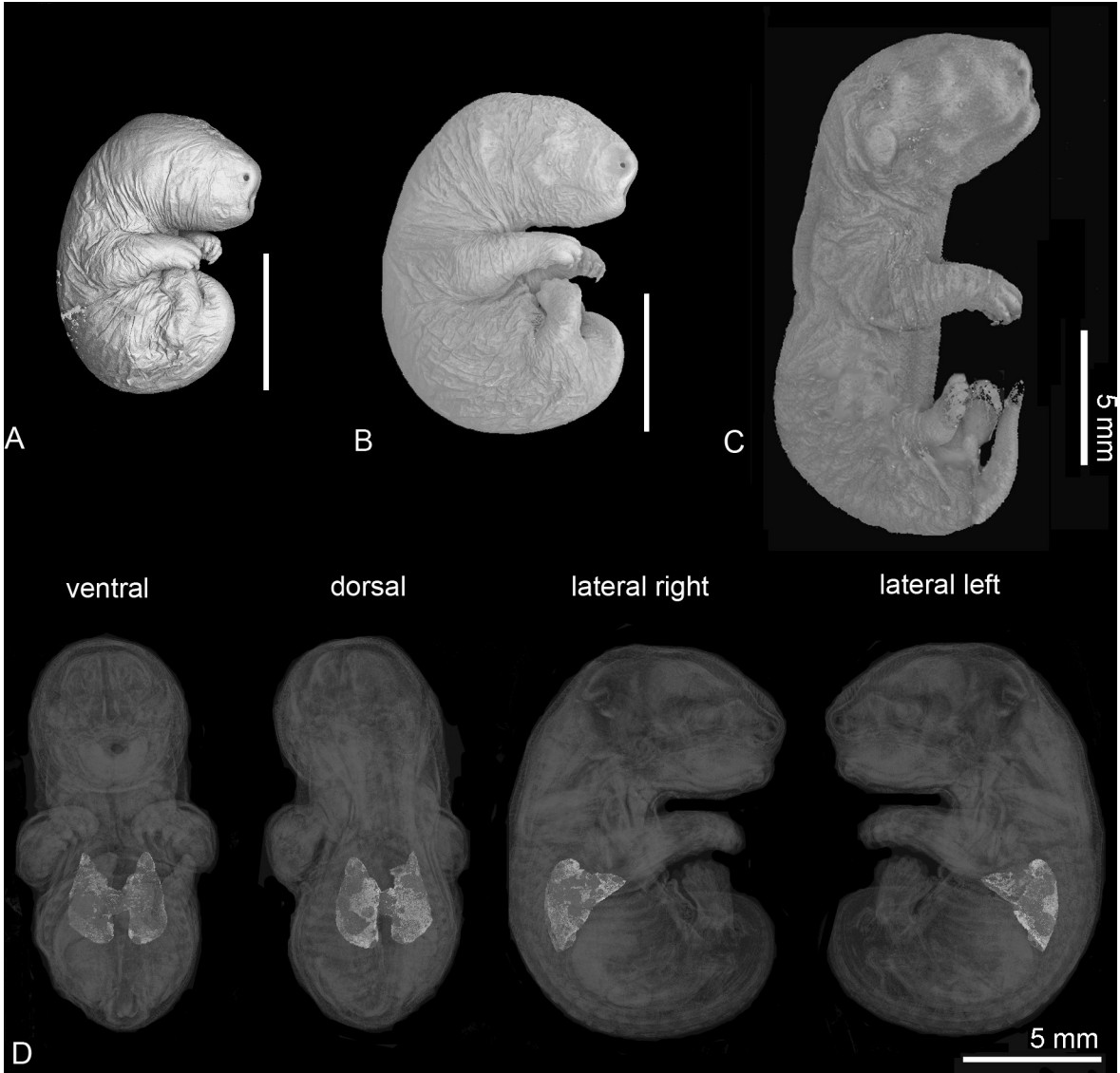

**Fig 1. External appearance of the gray short-tailed opossum.** Characteristic for marsupial offspring is the embryonic appearance, with strongly developed forelimbs and undifferentiated paddle-like hindlimbs as well as an undifferentiated oro-nasal region with oral shield in the first postnatal week (4 dpn (A), 7 dpn (B)). By 11 dpn (C) the ears, oral region and hind limbs appear more differentiated. Anatomical position of the lung from ventral, dorsal and lateral views (D, from left to right).

For better visualization of the functional lung units, selected terminal air spaces were segmented individually. Starting from the end of a terminal bronchiole the entire connected air space was segmented, creating a ROI for a single terminal air space. For the lungs at the alveolar period several terminal acini were segmented. To distinguish between adjacent terminal air spaces different colors (interval color) were chosen for the ROIs.

## Morphometric measurements

To quantify the structural development of the terminal air spaces, air space diameter and the thickness of the septa separating the air spaces were obtained using morphometric measurements. To ensure that selected lung sections are representative for the entire lung, all parts of

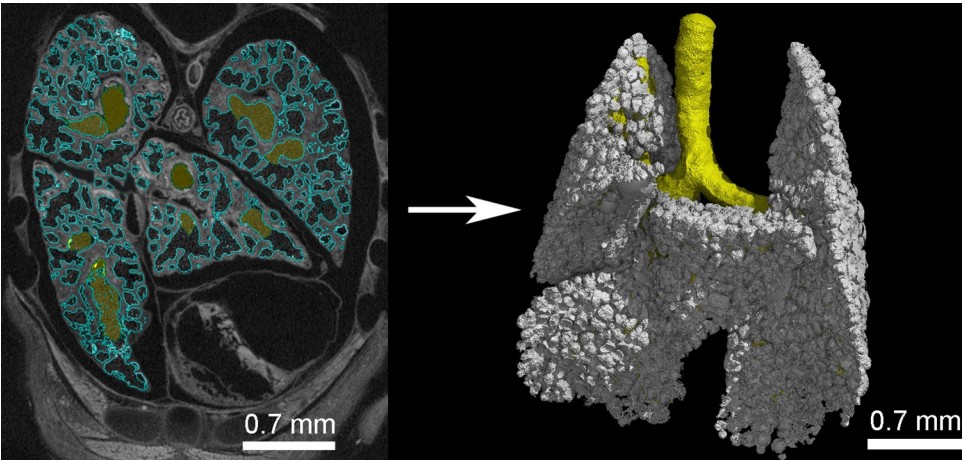

**Fig 2. Three-dimensional reconstruction of the lung of *Monodelphis domestica* at 7 dpn (2383_2) using μCT.** (A) μCT image in transverse section (2D) with marked air spaces (blue) and bronchial tree (yellow). (B) 3D reconstruction of the terminal air spaces (white) and the bronchial tree (yellow).

the lung should have equal probability of being sampled. This requirement is met by choosing a random starting point and employing uniform random sampling using the fractionator principal in a modified form [60]. The comprehensive sampling approach is based on serial sectioning through the entire lung, followed by systematic selection of a known fraction of the whole. The total length of the lung was estimated by the difference between start and end point after scrolling through the scanned lung. The total length of the lung divided by eight gave the sampling thickness for the fractionator. Following the fractionator eight digital pictures were taken from the 2D sections of the μCT-scans at the same magnification (ensured by the same scale) for each animal. Measurements were made directly on the computer screen using a digital ruler (ImageJ software; National Institutes of Health, USA) [61]. First the program was calibrated with the scale bar and a line was randomly cast over the image of the lung. On each digital photograph five air space diameter and five air space septa intersecting with the line were measured, yielding a total of 40 measurements for each lung. The values for single specimen are presented as mean with standard deviation, additional group means for the age stage (bold) are given (Table 1).

## Results

The volumes, surface areas, and morphological values reported in the results section are group means of all animals investigated for the respective age.

The lungs of the gray short-tailed opossum consist of six lung lobes, a cranial, a middle, a ventral and a caudal lobe in the right lung and a middle and a caudal lobe in the left lung. Fig 4 shows the ventral, dorsal, lateral, cranial and caudal views of the lung lobes in the newborn *Monodelphis domestica*.

The lungs of the near-term fetus at 13 dpc and in the neonate are at the canalicular stage of lung development and consist of large terminal air sacs, which open directly from the lobar bronchioles (Figs 5A, 5B, 6A–6C and 6D–6F). The terminal air spaces are deflated before birth, yielding a low lung volume of 0.53 mm$^3$. With birth the lungs become ventilated and the conducting airways and terminal air spaces are expanded by air, resulting in a lung volume of 2.15 mm$^3$ in the neonate (Table 1).

At this time the gas exchange takes place in the distal portions of the conducting airways, which are lined with respiratory epithelium, and the large terminal air spaces, which have a

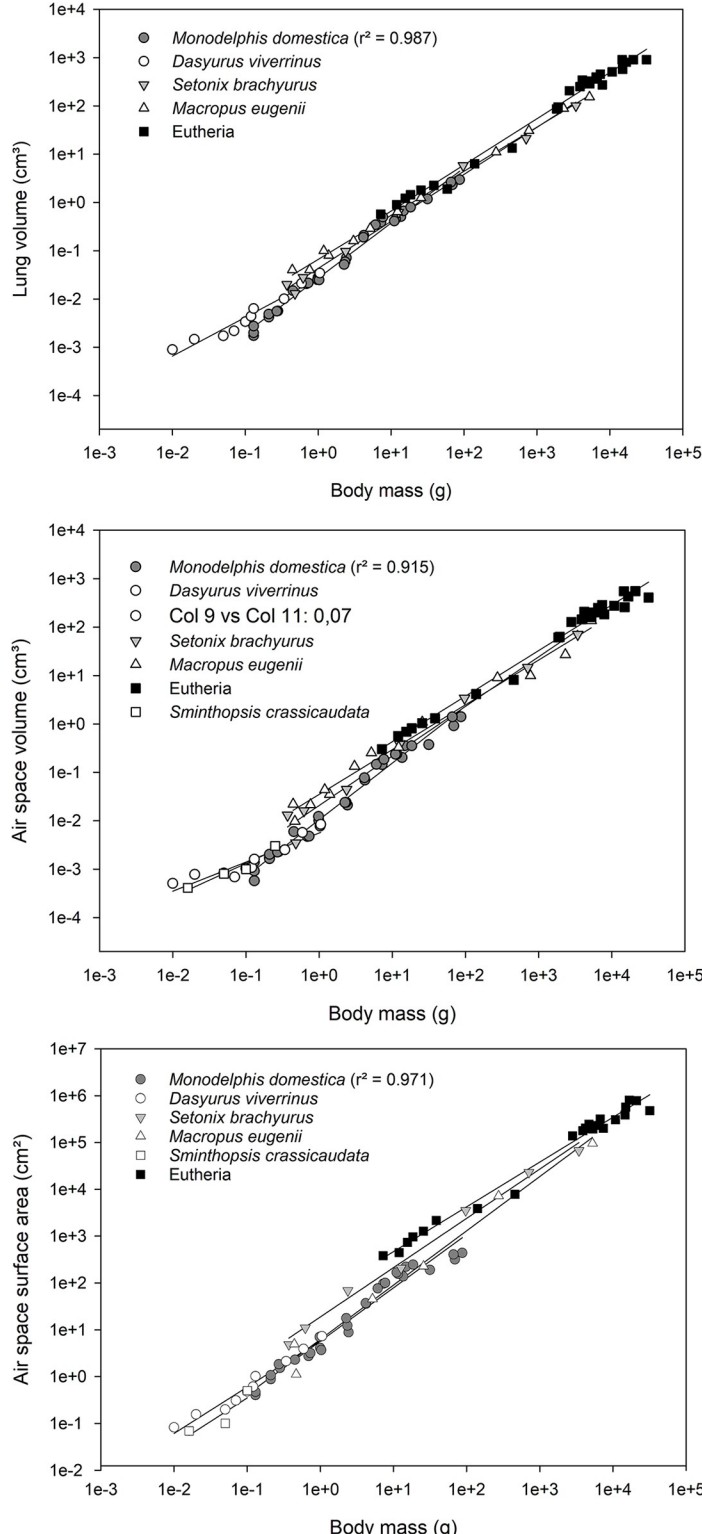

**Fig 3.** Double logarithmic plots of the lung volume (A), air space volume (B) and of the surface area (C) against body mass for *Monodelphis domestica* in the postnatal period. For comparison the plots include data from other marsupials: *Dasyurus viverrinus* [18]; *Sminthopsis crassicaudata* [16]; *Setonix brachyurus* [12–14] and *Macropus eugenii* [8,9,16,55,56] and eutherians: *Rattus norvegicus* [38]; *Bos taurus* [57]; *Sus scrofa domesticus* [58] and *Ovis aris* [59]. The graphs are based on individual animal data (eutherians are merged to on data set) and the regression lines are provided.

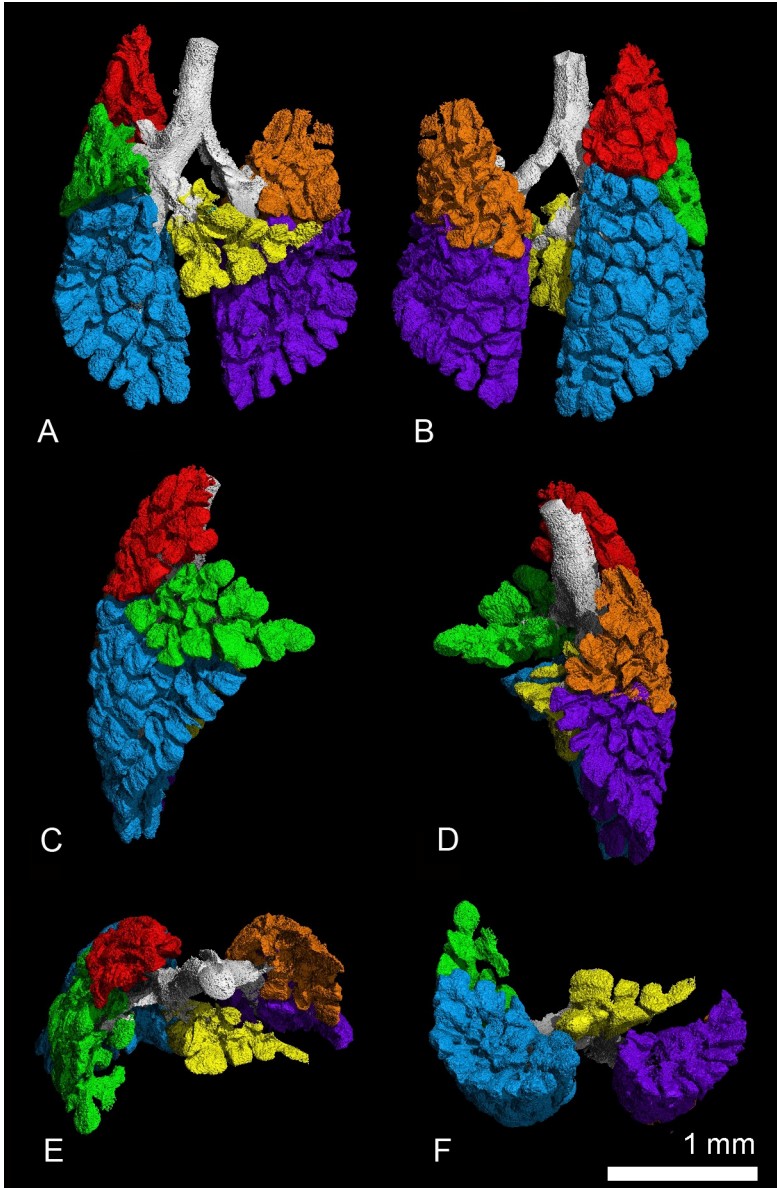

**Fig 4. Reconstruction of the terminal air spaces in the newborn lung of *Monodelphis domestica* with differentiation of the pulmonary lobes.** The lungs are shown in ventral (A), dorsal (B), lateral views from the right (C) and left (D) side and perspectives from the cranial (E) and caudal side (F). The pulmonary lobes are indicated by colors: right lung–cranial lobe (red), middle lobe (green), accessory lobe (yellow) and caudal lobe (blue); left lung–middle lobe (orange) and caudal lobe (purple).

lumen of 349 μm in diameter (Figs 6F and 7A). A thick interstitial layer (41 μm) separates the capillaries of one air space from the capillary network of the adjacent air space (Fig 7A).

The lung of a four days old *Monodelphis domestica* distinguishes from that of a newborn particularly through an increasing subdivision of the terminal air spaces (Figs 5C, 6G–6I and 7B). The terminal air sacs are still large with a diameter of 258 μm, but a number of septal ridges indicate a process of subdivision of the terminal air spaces. The septa separating the terminal air sacs decrease in thickness (34 μm).

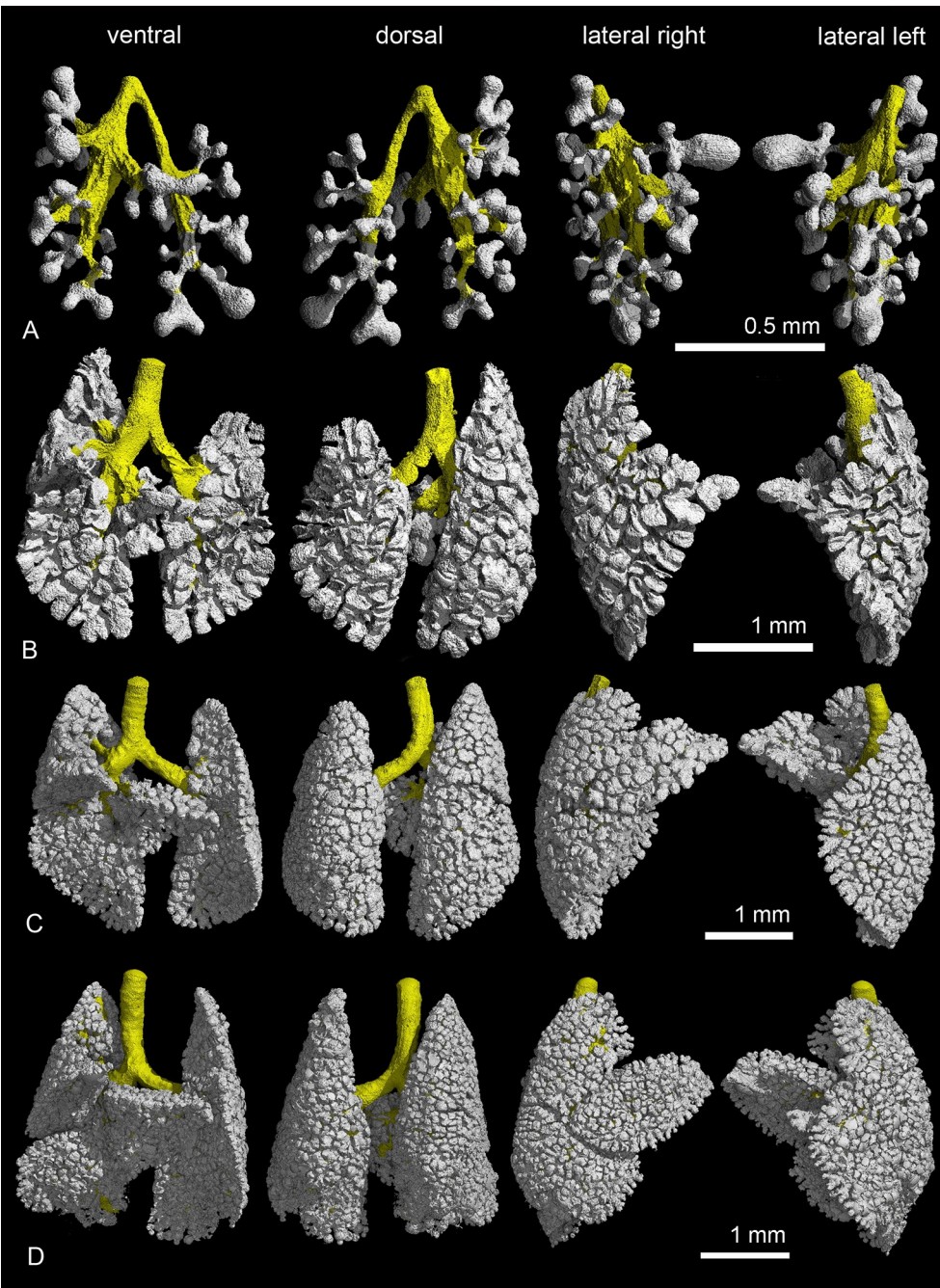

**Fig 5. Representative 3D reconstructions of the terminal air spaces of *Monodelphis domestica* during the first postnatal week.** Lung reconstructions at 13 dpc (A), in the neonate (B), at 4 dpn (C) and at 7 dpn (D). The lungs are shown from different perspectives: in ventral, dorsal and lateral views from the right and left side (from left to right).

By 7 dpn a further subdivision of the terminal air spaces and a gradually decrease in size of the terminal air sacs can be seen (Figs 5D, 6J-6L and 7C). A continuous double capillary bed, however with a thick interstitial layer, is present in the septa and indicates the transition to the saccular stage. The terminal air sacs have a size of 192 μm in diameter and septal crests that vary in height and thickness are numerous, resulting in an irregular shape.

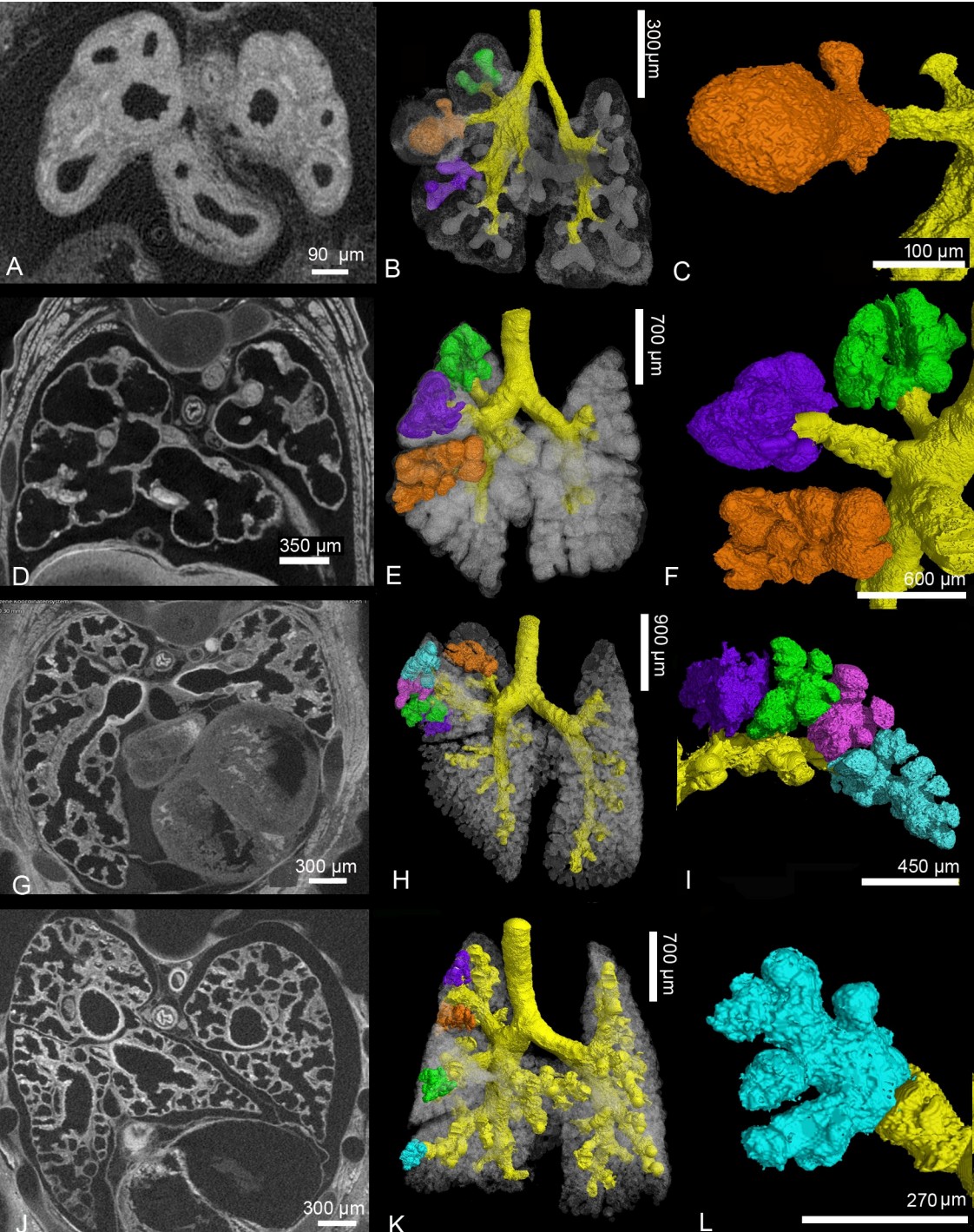

**Fig 6. Details of the developing *Monodelphis domestica* lung during the first postnatal week.** In embryos of 13 dpc (A-C) and in neonates (D-F) large terminal air spaces branch off directly from a simple bronchial tree, each forming a pulmonary lobe (F). By 4 dpn (G-I) the terminal air spaces become subdivided by septal growth. Branching from new formed segmental bronchioles several terminal air spaces can be distinguished in the pulmonary lobes (I). By 7 dpn (J-L) the sub segmentation of the air spaces progresses and the terminal saccules decrease in size (L). 2 D sections: A, D, G, J; position of terminal air spaces in the lung: B, E, H, K; close-up view of terminal air spaces: C, F, I, L.

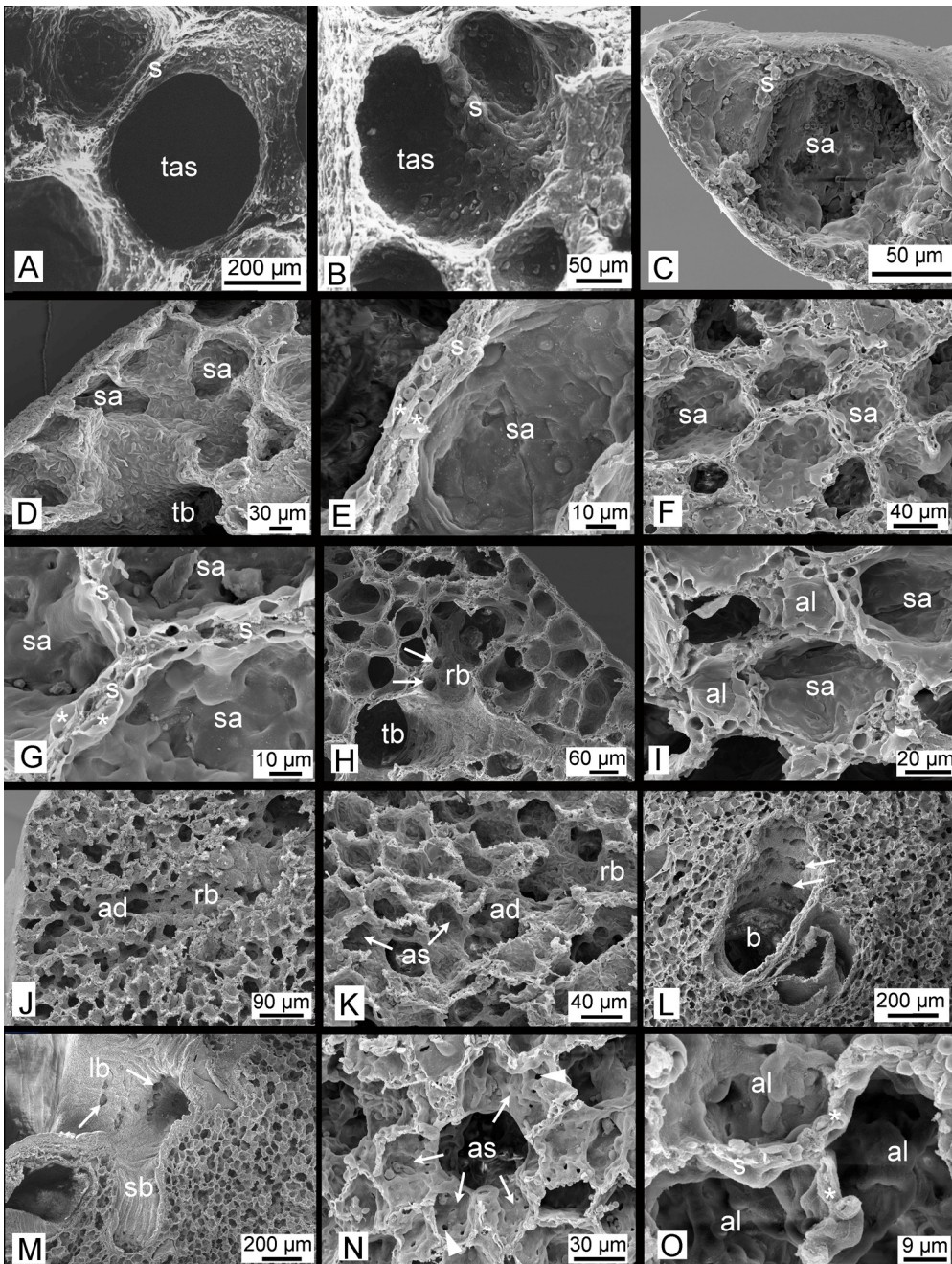

**Fig 7. Scanning electron micrographs of the developing *Monodelphis domestica* lung.** The lungs of neonates (A), at 4 dpn (B) and at 7 dpn (C) are characterized by large terminal air spaces, which become successively smaller with progressing subseptation. By 14 dpn (D, E) and 21 dpn (F, G) numerous small sacculi, which are separated from each other by thin double capillary septa (capillary beds are indicated by asterisks). By 28 dpn (H, I) first alveoli (indicated by arrows) can be seen and respiratory bronchioles develop. By 57 dpn (J-L) and in adults (M-O) a mature lung with respiratory bronchioles, alveolar ducts and alveolar sacs is present. Between alveoli single capillary septa have been formed. 'Pores of Kohn' (indicated by arrowheads) form interalveolar connections in walls of adjacent alveoli and connect them to each other. Magnification is indicated by the scale bar. ad, alveolar duct; al, alveolus; as, alveolar sac; b, bronchiole; lb, lobar bronchiole; rb, respiratory bronchiole; s, septum; sa, sacculus; sb, segmental bronchiole; tas, terminal air spaces; tb, terminal bronchiole.

Further compartmentalization occurs in the lung between 11 and 21 dpn (Fig 8A–8C). Substantial changes take place in the architecture of the lung (Fig 9A–9L). The terminal saccules become more numerous and decrease in size (Fig 7D and 7F). They measure 141 μm in diameter by 11 dpn, 126 μm by 14 dpn and 108 μm by 21 dpn. Several new saccules develop near the pleura. They are separated by septa standing vertically on the pleura (Fig 9K). The saccules are still separated by a double capillary septum (Fig 7E and 7G). However, the septa decrease in thickness and measure 18 μm by 21 dpn. In contrast to earlier stages a thick core of stromal cells with large interstitial spaces is missing (Fig 7G). The lung volume and surface area increase through an extensive rise in saccular number, as well as structural complexity (Table 1, Fig 7H).

With 28 dpn the transition from the saccular to the alveolar period of lung development starts. The terminal air spaces, consisting of saccules and newly formed alveoli, are characterized by a further increase in number and decrease in size (Fig 8D and 9M–9P). Saccules, characterized by a double capillary septum, still dominate the lung parenchyma by 28 dpn. However, a few small alveoli, separated by a single capillary septum, are already present (Fig 7I). The terminal air spaces measure 89 μm in diameter and the thickness of the septa separating the air spaces is with 20 μm comparable to 21 dpn. With the formation of alveoli first respiratory bronchioles develop. They are characterized by flattened epithelium with alveoli located in their walls (Fig 7H). The further rise in saccular number and new formation of alveoli lead to an increase in lung volume, air space volume and surface area by 28 dpn (Table 1).

The lung at 35 dpn is characterized by a further increase in lung volume and the proceeding formation of alveoli (Table 1, Figs 10A and 11A–11D). The lung has fully attained the alveolar stage. The bulk of alveoli lead to a further increase in air space volume and surface area (Table 1). The terminal air spaces are dominated by alveoli and measure 70 μm in diameter. The septa between the air spaces are predominantly single capillary septa and measure 16 μm in thickness. Respiratory bronchioles with alveoli are found more frequently compared to 28 dpn. The distal parts of the respiratory bronchioles pass into alveolar ducts which open into alveolar sacs. Thus, typical structures of the mammalian acinus are present.

By 49 dpn (Fig 10B and 11E–11H) no distinct structural changes can be seen in the terminal air spaces of *Monodelphis domestica*. The terminal air spaces further decrease in size (52 μm) and the air space septa decrease in thickness (12 μm). In the 57 days old lung (Fig 10C and 11I–11L) alveoli have markedly increased in number. With the progressing formation of alveoli, numerous respiratory bronchioles can be seen (Fig 7J). The distal parts of the respiratory bronchioles pass into short alveolar ducts, which are covered with respiratory epithelium and have alveoli at their sides. The alveolar ducts open into alveolar sacs, from which alveoli radiate into the surrounding parenchyma (Fig 7K). The size of terminal air spaces (51 μm) and the thickness of the single capillary septa (10 μm) are similar to that of 49 dpn. In addition to this regular development, alveoli can be found also at the walls of solely conducting airways, such as segmental and terminal bronchioles. The bronchial walls are perforated by the openings of numerous alveoli (Fig 7L).

The lung parenchyma of the adult *Monodelphis domestica* is strongly subdivided and the gas exchange surface area has markedly increased (Table 1, Figs 10D and 11M–11P). Shortly after branching off from the lobar bronchi, the walls of the segmental bronchi are perforated by numerous alveoli (Fig 7M). These irregular formations of alveoli, similar to that described at 57 dpn, are widespread within the adult lung. An alveolar acinus, typical for the adult mammalian lung, is present. Alveolar sacs contain multiple alveoli, which radiate like a raspberry from the center (Fig 7N). The alveoli measure 79 μm in diameter. Compared to 49 and 57 dpn the terminal air spaces appear to be expanded. In the interalveolar septa pores of Kohn are numerous and present throughout the adult lung. The pores of Kohn (Fig 7N, arrowheads)

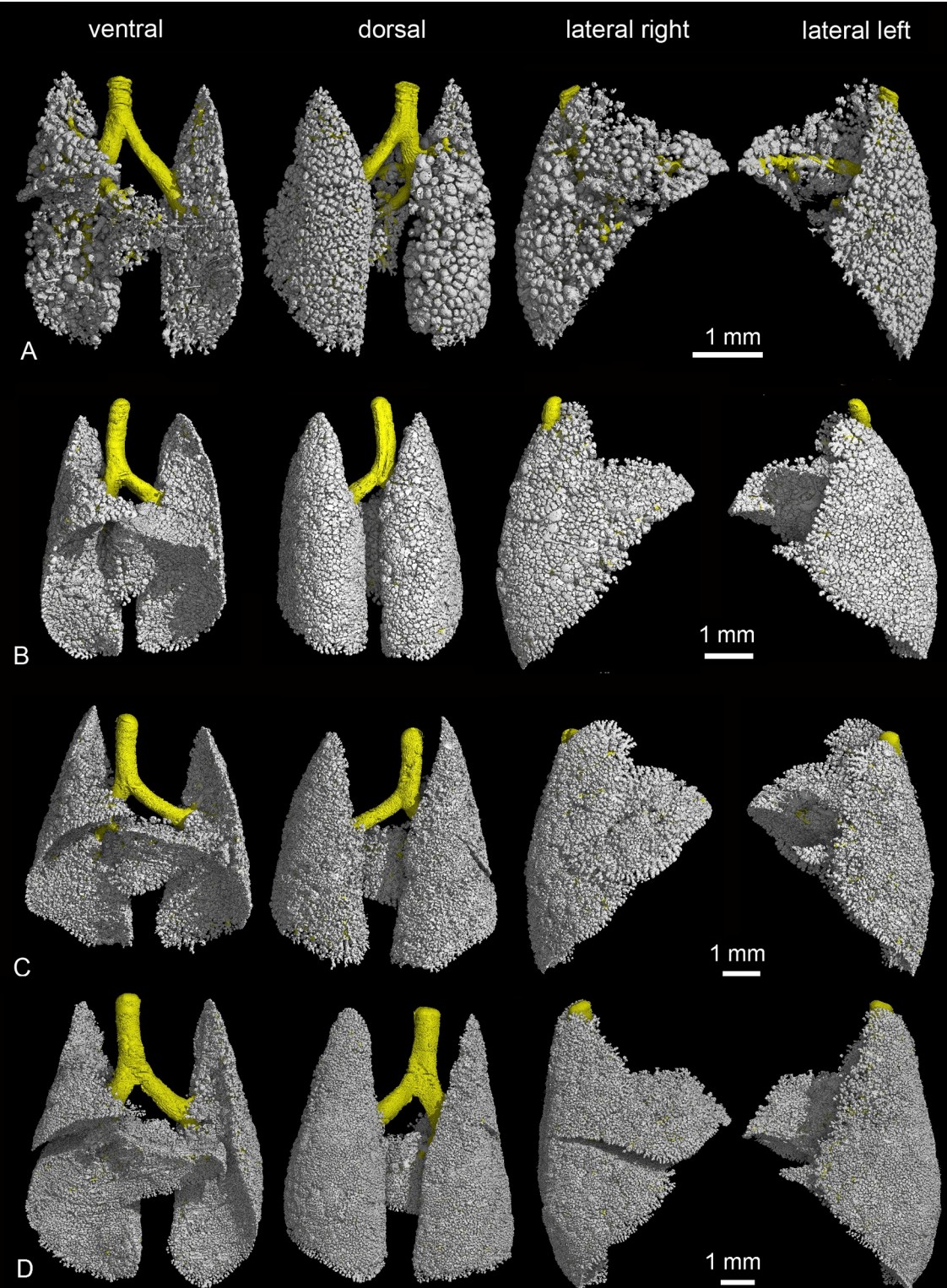

**Fig 8. Representative 3D reconstructions of the terminal air spaces of *Monodelphis domestica* from 11 to 28 postnatal days.** Lung reconstructions at 11 dpn (A), 14 dpn (B), 21 dpn (C) and at 28 dpn (D). The lungs are shown from different perspectives: in ventral, dorsal and lateral views from the right and left side (from left to right).

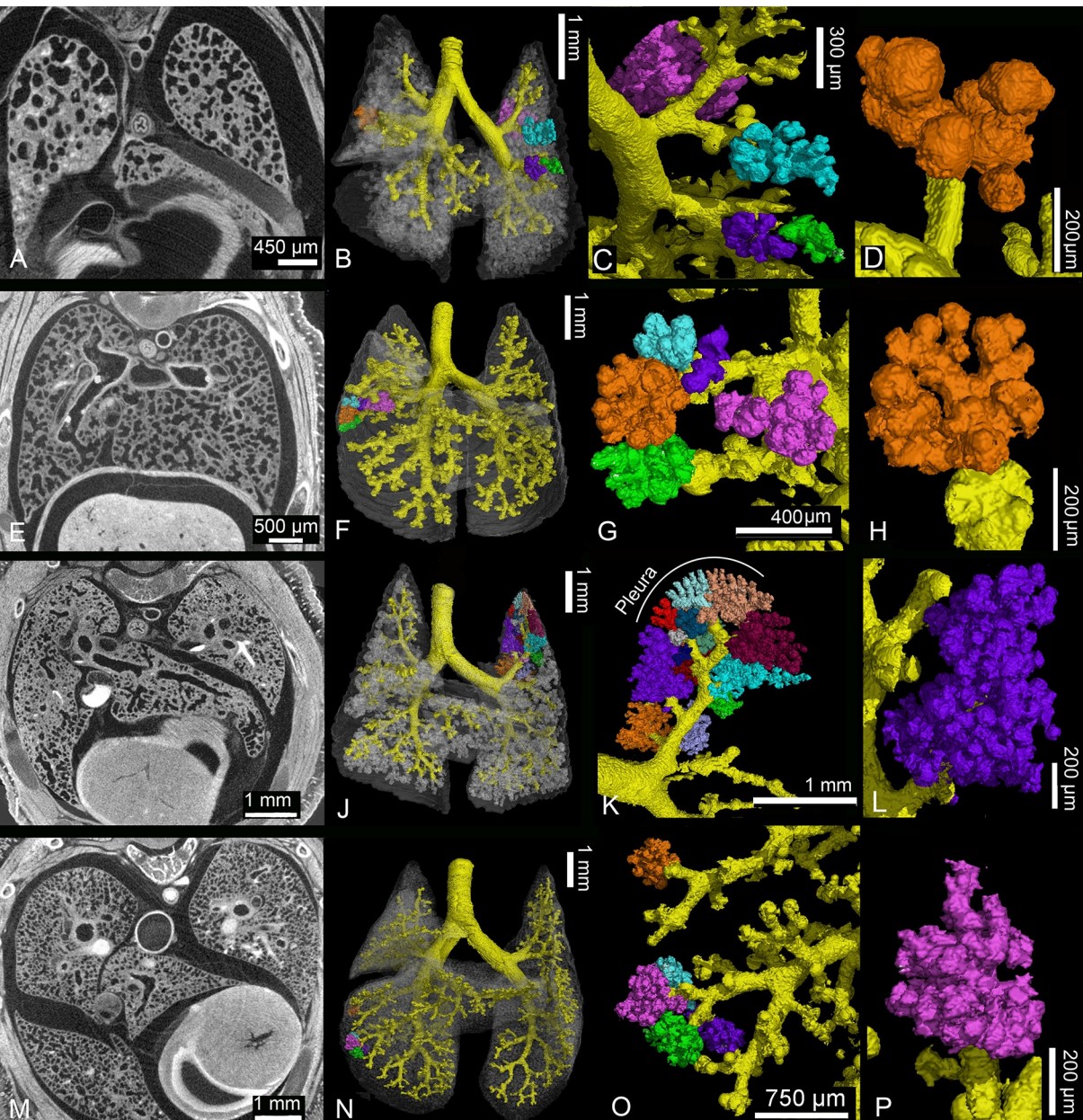

**Fig 9. Details of the developing *Monodelphis domestica* lung from 11 to 28 postnatal days.** Compartmentalization of the terminal air spaces progresses at 11 (A-D), 14 (E-H) and 21 (I-L) dpn. The terminal saccules become more numerous and decrease in size. Several new saccules develop near the pleura, separated by septa vertically standing on the pleura (K, red, bright blue and beige). By 28 dpn (M-P) alveolarization starts. The terminal air spaces consist of saccules and new formed alveoli. 2 D sections: A, E, I, M; position of terminal air spaces in the lung: B, F, J, N; terminal air spaces: C, G, K, O; close-up view of terminal air space: D, H, L, P.

form a connection between adjacent alveoli. The single capillary septa separating the alveoli measure 10 μm in width. A centrally located capillary occupies the septum almost entirely (Fig 7O).

Lung volumes ($V_L$) and terminal air space volumes ($V_A$) in relation to body mass for all specimens examined from neonate to adult are presented in Fig 3A and 3B. From neonate to adult a steady increase in lung and air space volume could be observed. Over all, developmental stages, $V_L$ (r = 0.987) and $V_A$ (r = 0.915) are closely correlated to body mass.

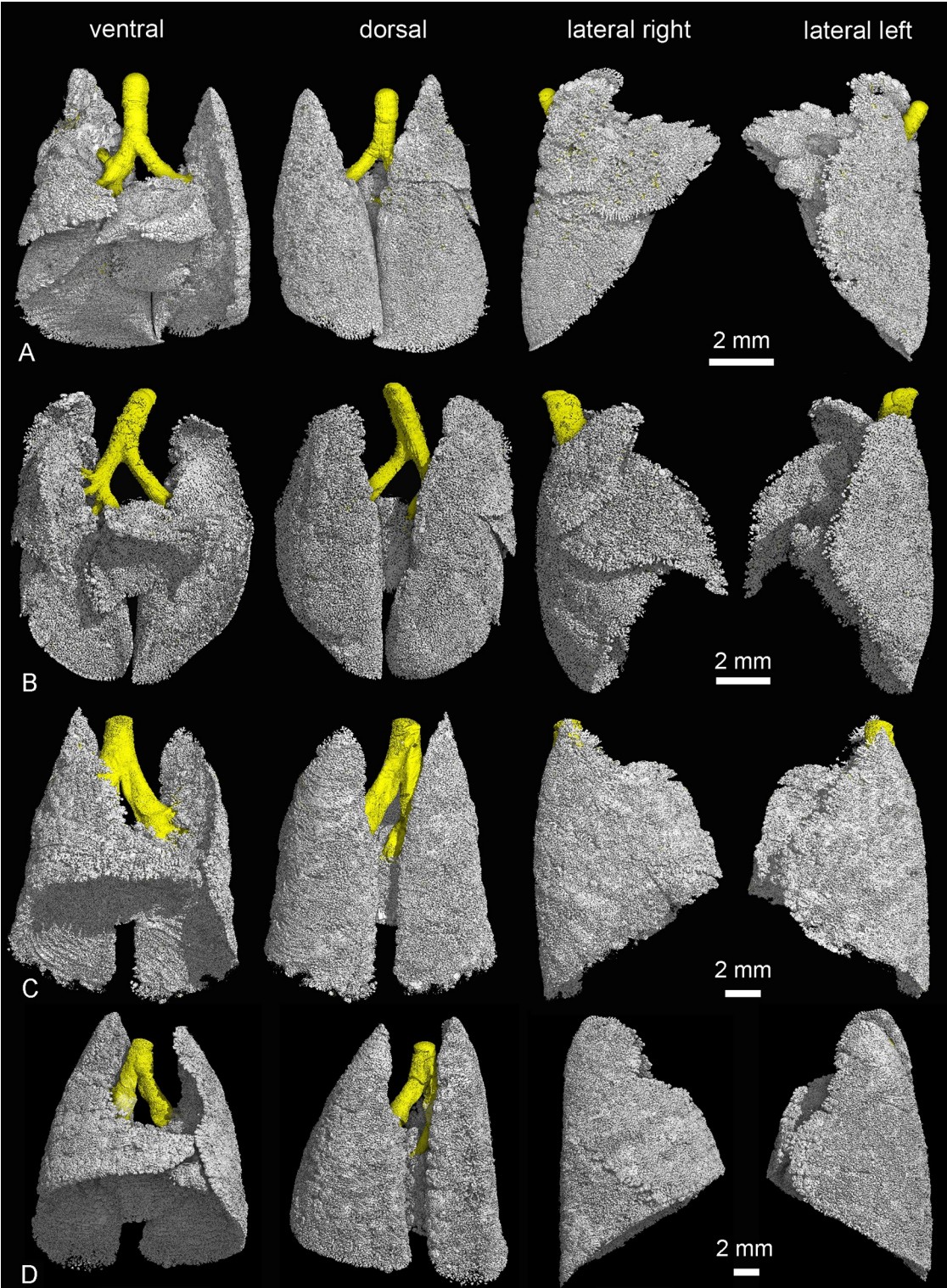

**Fig 10. Representative 3D reconstructions of the terminal air spaces of *Monodelphis domestica* from 35 to 57 postnatal days and in adults.** Lung reconstructions at 35 dpn (A), 49 dpn (B), 57 dpn (C) and in an adult (D). The lungs are shown from different perspectives: in ventral, dorsal and lateral views from the right and left side (from left to right).

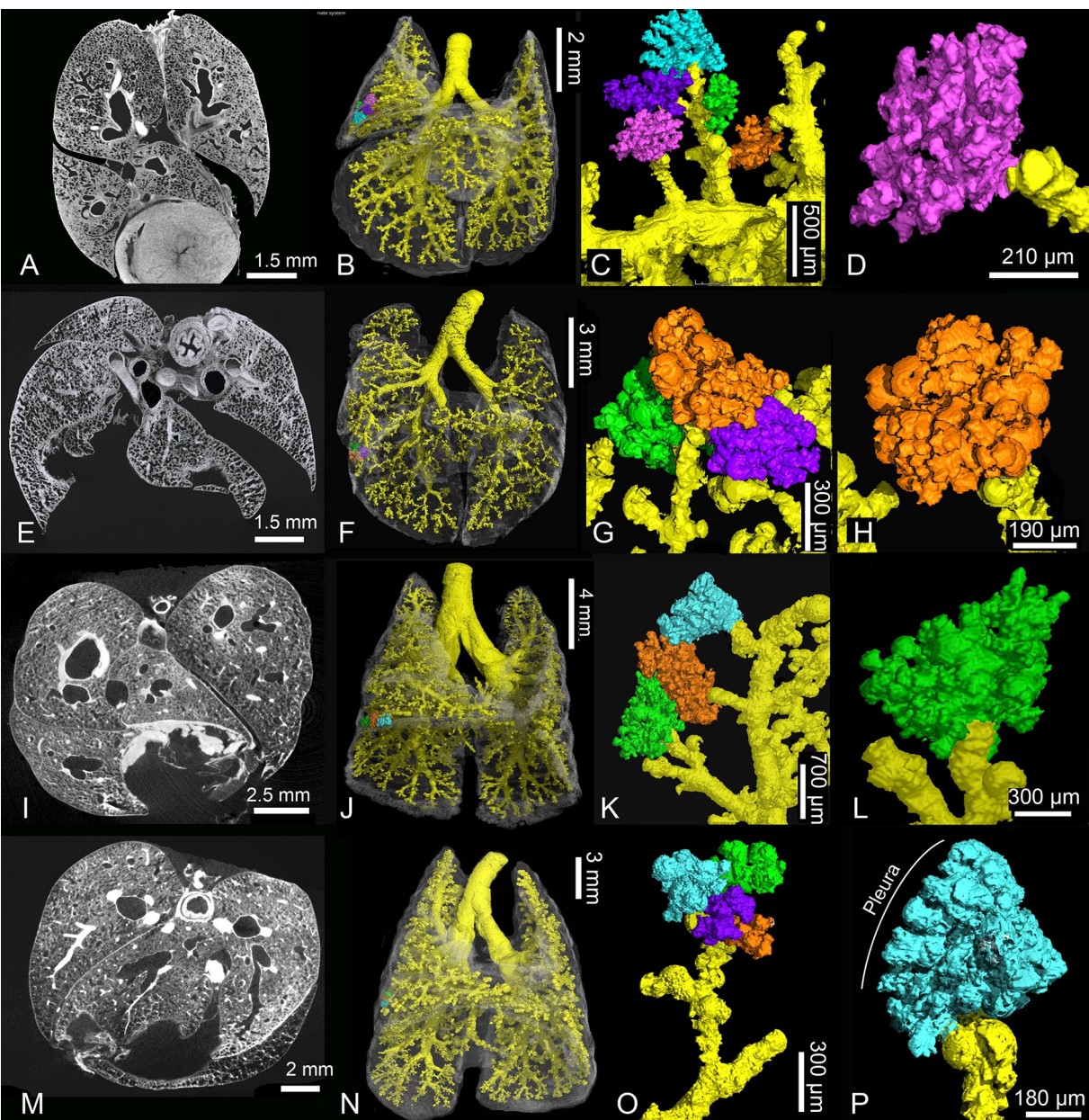

**Fig 11. Details of the developing *Monodelphis domestica* lung from 35 to 57 postnatal days and in adults.** By 35 dpn (A-D) the lung has fully attained the alveolar stage. The terminal airways consist of respiratory bronchioles and alveolar ducts. The lung parenchyma of *Monodelphis domestica* at 49 dpn (E-H), 57 dpn (I-L) and in adult (M-P) is strongly subdivided and the alveoli in their entirety provide a large surface area for gas exchange. Alveoli located at the pleural surface are separated by septa vertically standing on the pleura (P) similar as observed in sacculi close to the pleural surface by 21 dpn. 2 D sections: A, E, I, M; position of terminal air spaces in the lung: B, F, J, N; terminal air spaces: C, G, K, O; close-up view of terminal air space: D, H, L, P.

The surface area ($S_A$) of the terminal air spaces in relation to body weight (Fig 3C) indicates a continuous increase of the gas exchange area during the structural transformation of the lung in the postnatal period. With progressing compartmentalization of the lung, the airspace surface area increases steadily. The airspace surface area is positively correlated with body mass (r = 0.971).

## Discussion

With birth, the lungs of newborn mammals have to take over the function of gas exchange, formerly provided by the placenta. Viability of the neonate depends on an adequately developed respiratory system [21,16].

The lungs of newborn marsupials are not fully developed at birth, as they are born in a relatively immature state compared to placental mammals. Therefore, cutaneous respiration supports gas exchange to various amounts depending on the degree of maturity of the lung [21,23,24,26].

The lung structure of marsupial neonates follows the size variation in the sequence G1 to G3 [62]. A gradation of lung development from early canalicular stage (G1), late canalicular (G2) to early saccular stage (G3) can be observed among newborn marsupials [26,49]. The newborn gray short-tailed opossum has large terminal air spaces. The cranial, middle and accessory lobes of the right lung consist of one large terminal air space respectively. The septum consists of capillaries on both sides, forming a blood-air-barrier facilitating gas exchange [15]. However, a continuous double capillary septum is not present yet, attributing the lung of the newborn gray short-tailed opossum to the late canalicular stage (G2) [17]. A similar lung structure at birth can be seen in the Virginia opossum [4], the brushtail possum [7] and the bandicoot [5,6]. The most immature lungs among marsupial neonates are present in dasyurids (G1) like the eastern and northern native cat [3,7,18] and the stripe-faced and fat-tailed dunnart [16,63,64]. They have lungs consisting of well vascularized air chambers that appear like two 'air bubbles'. More developed lungs (G3) than those of the newborn gray short-tailed opossum can be found in kangaroos [8,9,11–14] and the koala [26]. These lungs consist of a primitive bronchial tree terminating in several saccules.

The compartmentalization of the lung of the gray short-tailed opossum progresses fast during the first postnatal weeks. The terminal air spaces become more and more subdivided by the formation of new double capillary septa from existing septa. By 7 dpn a primary septum with a continuous double capillary bed is present, indicating that the lung has entered the saccular stage. The terminal sacculi become smaller with time and the newly formed septa get thinner. The saccular stage can be characterized by the formation of transitory saccules, which are progressively subdivided by septation into more generations of saccules [18]. The process of saccule multiplication is very similar to that of alveolar formation [14,39]. However, in contrast to the alveolar stage, microvascular maturation, a process that leads to the formation of mature septa with a single capillary layer, does not occur during sacculation.

A prolonged period of saccular subseptation has been described also for the Virginia opossum [4], the tammar wallaby [8,9], the bandicoot [6] and the quokka [11,14].

Development, from the terminal air sac stage to an alveolar lung, takes place over an extended period of > 180 days in the tammar wallaby [9]. During this long saccular stage, periods of tissue proliferation and periods of expansion alternate. Up to 20 dpn, volume increases of the terminal air spaces are largely due to expansion of the air spaces. Around 30 dpn, rapid tissue proliferation and development of septa which subdivide the air spaces take place. This burst of tissue proliferation is followed by a period during which growth is again largely due to expansion. Between 70 and 180 dpn subdivision of the large air sacs into much smaller compartments occurs again as the result of tissue proliferation and septal development [8]. For the rat a similar differentiation of periods of septa formation and expansion of terminal air spaces have been described during alveolarization [42], indicating analogies in saccular and alveolar formation.

In the gray short-tailed opossum and most other marsupial species, the saccular stage is much shorter. Alternating periods of tissue proliferation and expansion as described for the

tammar wallaby are not detectable. Sacculation in the gray short-tailed opossum appears to be a continuous process with tissue proliferation and expansion of air spaces taking place in parallel. The first mature single capillary septa, indicating the onset of alveolarization, can be seen at 28 dpn and the full alveolar stage is attained at 35 dpn in *Monodelphis domestica*.

Similar times for the onset of alveolarization have been reported for the fat-tailed dunnart (45 dpn) [63], the brushtail possum (39 dpn) [65], the bandicoot (40 dpn) [5,6] and the Virginia opossum (45 dpn) [4].

In contrast to marsupials, the lung of newborn placental mammals is either at the late saccular stage in altricial species or at the alveolar stage in precocial species [44,49]. Postnatal lung development in altricial placentals is rapid and the formation of alveoli starts in the golden hamster and mouse at the age of 2 dpn [15,30] and in the musk shrew and the rat at 4 dpn [15,39]. The majority of placental mammals, among them all precocial species, reach the alveolar stage in utero and possess alveoli already at birth [44,49]. The entirety of the numerous small alveoli provides a large gas exchange surface area, a necessity to meet the high metabolic demands accompanying precociality.

The growth of alveoli, including additional formation of new units, proceeds in the postnatal period in marsupial and placental mammals. In marsupials the process of alveolarization continuous until 85 dpn in the Virginia opossum [4], until 113 dpn in the brushtail possum [65], until 125 dpn in the quokka [14] and until 180 dpn in the tammar wallaby [8].

For long time it was assumed that alveolarization ceases after the capillary layers in the alveolar septa mature during microvascular maturation [39]. However, new studies using recently developed techniques report continued alveolarization in placental mammals for rabbits [66], rats (until 60 dpn) [43,67], and mice (until 36 dpn) [30,42]). Alveolarization continues at least until young adulthood in the rhesus monkey [68] and in humans [69,70]. The potential for alveolarization might be preserved throughout life allowing regeneration from degenerative lung diseases [32,71].

New alveolar septa may be lifted off both immature and mature alveolar septa, allowing new septa or new alveoli to be formed principally at any time, even during adulthood [32]. Two phases of alveolarization are distinguished: classical (or bulk) alveolarization and continued alveolarization [42,43]. The process of alveolarization has been described in detail by [32]. The primary septa present at the early saccular stage in marsupials contain a double-layered immature capillary network. At sites where new septa (or secondary septa) will be formed, smooth muscle cell precursors, elastic fibers and collagen fibrils accumulate. The new septa form by an upfolding of one of the two capillary layers. The resulting newly formed secondary septa possess two capillary layers and subdivide preexisting air spaces into smaller sacculi. This process continues as sacculation until the first alveoli are formed. During microvascular maturation, the double-layered capillary network fuses to a single-layered one. During continued alveolarization, new alveolar septa are still formed by an upfolding of the capillary layer, even if the alveolar surface opposing the upfolding is now missing its capillaries. In all modes (sacculation, classical alveolarization and continued alveolarization) a sheetlike capillary layer folds up to form a double-layered capillary network inside the newly formed septum. When a new alveolar septum is formed, it will mature shortly after by a fusion of the double-layered capillary network [32].

A recent 3D study of the mouse lung suggests that alveolar "septal tips" are in fact ring or purse string structures containing elastin and collagen [72]. Saccular formation and later alveolarization in the terminal air spaces seem to take place by epithelial extrusion through a directionally orientated orifice with a ring or purse string-ring lip that imparts some localized stiffness to the mesenchyme. The alveolar epithelium extrudes outwards into the surrounding

mesenchyme, which is correspondingly less stiff than the saccular or alveolar orifice, and forms 'bubble' like structures, the sacculi or alveoli [72].

At birth, the gray short-tailed opossum weighs approximately 126 mg and has a $V_L$ of only 2.15 mm$^3$. Adult females have body weights of 74 g and a $V_L$ of 2,629.33 mm$^3$. This means that from birth to adulthood the body weight increases around 550 times and the $V_L$ increase is 1,300-fold. In larger marsupial species an even higher increase of lung volume has been reported, e.g., 3,800 times in the tammar wallaby [9] and 8,000 times in the quokka wallaby [12]. In placental mammals the $V_L$ increase from birth to adulthood is much lower, only approximately 23 times in humans and rats [73].

Low lung volumes, similar to that of the newborn gray short-tailed opossum, were reported for other marsupial neonates, e.g., 4–9.7 mm$^3$ in the tammar wallaby (body weight 0.465 g) [9,16], 2 mm$^3$ in the quokka (0.370 g) [12,13], 0.9 mm$^3$ in the eastern quoll (0.0125 g) [18] and 0.4 mm$^3$ in the fat-tailed dunnart (0.010 g) [16]. For comparison the $V_L$ of the newborn rat (7.2 g), a typical altricial placental, is with 570 mm$^3$ much higher [13]. The lung volumes determined in newborn marsupials are lower than that predicted from allometric equations for placental neonates [21]. The low lung volume at birth in marsupials may relate to the stage of lung development; the earlier the stage the lower the volume and the greater the extent of cutaneous gas exchange [16].

The volumes of the lung and of the terminal air spaces and the air space surface area increased steadily during the postnatal period in *Monodelphis domestica* (Table 1). Overall, the developmental stages, $V_L$, $V_A$ and $S_A$ were closely correlated with body mass (Fig 3). The absolute air space surface area of the newborn lung (0.433 cm$^2$) increased ~10-fold by 14 dpn (4.923 cm$^2$). In the more immature born eastern quoll the surface area of the air spaces increased from 0.028 cm$^2$ to 2.122 cm$^2$ even 76-fold during this time period [18]. The surface areas of the lung available for gas exchange in the newborn tammar wallaby (1.115 cm$^2$) and dunnart (0.069 cm$^2$), determined by phase contrast X-ray imaging [16], are comparable to the results of this study. Similar to $V_L$, the surface areas reported for marsupials are below values predicted from allometry for eutherian species [16,21].

Several studies examined morphometric aspects of the developing lung in marsupials [9,12,14]. In the tammar wallaby, studies have shown that changes in the surface area of the lung up to 20 dpn are largely due to expansion of the air spaces while tissue proliferation and air sac subdivision is most pronounced during the transitional period from ectothermy to endothermy (after 70 dpn,100 g) [9]. However, in smaller marsupial species, like the gray short-tailed opossum tissue proliferation and air sac subdivision can be observed already in the early postnatal period and proceeds obviously faster than in the tammar wallaby. A marked increase of air space surface area could be seen in the gray short-tailed opossum between 14 dpn and 28 dpn, probably due to massive septal development and expansion of existing and new formed sacculi during the late saccular stage. Between 28 dpn and 35 dpn, at the time alveolarization took place, air space surface area more than doubled. In marsupial postnatal development, the airspace surface areas show highest rates of increase during the alveolar stage [12,14]. However, this correlation of increase in surface area and alveolarization seems to be typical for mammalian lung development in general. Also, morphometric studies in placental species, like the rat, found that the formation of alveoli, occurring between 4 and 10 dpn, results in an increase of gas exchange surface area by a factor of 2.6, whereas the lung volume increased only by a factor 1.6 during this period [38,39,74].

The lung volumes reported in this study might differ from the functional lung volumes of living animals. Limitations due to the method of fixation need to be discussed. Specimens from 0 to 28 dpn were totally immersed in fixative after severing the head to allow fixative inflow. This might result in lower lung volumes than the functional value, because lungs tend

to deflate below functional residual capacity (FRC) at death [16]. The inflation degree of the lung cannot be controlled with that method, causing a higher variation in lung volumes. In older stages (35 dpn–adult) the lungs were filled with fixative to a tracheal pressure of 20 cm. That may lead to an overestimation of lung volume and surface area since lungs inflated with liquid have a larger compliance and are easier to distend than air-filled lungs [75]. Regardless of the preservation methods, the lung volumes and surface areas calculated for *Monodelphis domestica* are comparable to values reported for other marsupial pouch young of similar size (see Fig 3). In addition, others have demonstrated the comparability of lung volumes and surface areas derived from computed tomography data sets with histological estimations [76].

## Conclusion

Microcomputed tomography (μCT) offers the possibility to show the comprehensive structural transformations in the developing lung of the gray short-tailed opossum in 3D. The results confirm that marsupials such as the gray short-tailed opossum are born with structurally immature lungs when compared to eutherian mammals and undergo a marked increases in architectural complexity during the postnatal period. In marsupials, the process of alveolarization, which takes place largely intrauterine in the eutherian fetus, is shifted to the postnatal period and is therefore more easily accessible for investigation. The structural development of the terminal air spaces from large terminal sacs to the final alveolar lung takes place in functional state in a continuous morphogenetic process. This allows insights in the structural prerequisites of a functioning lung and opens a new window for better understanding of the evolution of mammalian lung development. It can be assumed that mammalian lung development follows similar developmental pathways in all mammalian species, including marsupials.

## Acknowledgments

I am very grateful to Kristin Mahlow (lab manager of CT-lab) for staining, sample preparation and acquisition of μCT-scans. I thank the animal keepers Petra Grimm and Annett Billepp as well as Dr. Peter Giere and Dr. Peter Bartsch of the animal facility of the Museum für Naturkunde Berlin for breeding and providing the animals for this project. Furthermore, I would like to thank the referees for their inspiring comments, which improved the clarity of the study.

## Author Contributions

**Conceptualization:** Kirsten Ferner.

**Funding acquisition:** Kirsten Ferner.

**Investigation:** Kirsten Ferner.

**Methodology:** Kirsten Ferner.

**Project administration:** Kirsten Ferner.

**Supervision:** Kirsten Ferner.

**Validation:** Kirsten Ferner.

**Visualization:** Kirsten Ferner.

**Writing – original draft:** Kirsten Ferner.

**Writing – review & editing:** Kirsten Ferner.

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
