## [Decision Letter · Decision Letter 0]

9 Nov 2023

PONE-D-23-30643Development of the terminal airspaces in the Gray short-tailed opossum (<monodelphis domestica="">) – 3 D reconstruction by microcomputed tomography</monodelphis>PLOS ONE

Dear Dr. Ferner,

Thank you for submitting your manuscript to PLOS ONE. After careful consideration, we feel that it has merit but does not fully meet PLOS ONE’s publication criteria as it currently stands. Therefore, we invite you to submit a revised version of the manuscript that addresses the points raised during the review process.

We look forward to receiving your revised manuscript.

Kind regards,

Josué Sznitman

Academic Editor

PLOS ONE

Journal Requirements:

Reviewers' comments:

Reviewer's Responses to Questions

**Comments to the Author**

1. Is the manuscript technically sound, and do the data support the conclusions?

Reviewer #1: Partly

Reviewer #2: Yes

2. Has the statistical analysis been performed appropriately and rigorously? 

Reviewer #1: N/A

Reviewer #2: N/A

3. Have the authors made all data underlying the findings in their manuscript fully available?

Reviewer #1: No

Reviewer #2: No

4. Is the manuscript presented in an intelligible fashion and written in standard English?

Reviewer #1: Yes

Reviewer #2: Yes

5. Review Comments to the Author

Reviewer #1: The author addresses the lung development of the Gray short-tailed opossum with focus on the terminal airspaces. The study provides good context across different species and lists and discusses similarities and differences. State of the art µCT is employed to create 3D volume images of the lungs. Samples from a great number of time points in the development are examined.

The main value of this study lies in the qualitative description of lung development over a high number of time points. This shows the distinct of development steps of the lungs. Appropriate images are provided to display the developmental differences.

The qualitative description is extended by a quantitative assessment of the lungs. The low sample sizes for each group and uncontrolled variables such as lung inflation, different preparation protocols etc. means this is in no way a representative numerical description for each time step. Nevertheless, it serves as an orientation and is a great point were future studies, providing more samples for individual development steps, might extend the knowledge gathered in this initial study.

A detailed discussion of developmental processes concludes this work.

My main issue with this paper concerns its reproducibility and comparability with current and future studies. To aid this, I would wish for a more detailed method description. This could turn this paper into a great foundation on which many future works on the Gray short-tailed opossum lung might be based.

Other than that, I have only some minor remarks on areas were slight improvements might aid the presentation of the results. A detailed list of all my recommendations is attached below.

Major points

-P7 L8f What results were acquired using these eight animals? As they were said to have been studied using SEM I guess they might have been used to measure diameter or thickness of morphological structures. This should be described in the methods section (see below).

I looked into the cited PhD thesis and was able to piece together from Table I, by the provided ages and the fact that SEM was performed on them, which lung sections were looked at. Data that would allow the reader to compare these animals to the 35 listed in the current study, e.g. body weight, was not reasonably available.

-P9 L15 How was the reconstruction performed? Manual tracing? Thresholding? Watershed segmentation? AI pattern recognition? Please specify.

-P9 L16 What did the ROIs consist of? What was included, what excluded and why? How were e.g. extrapulmonary airways or extrapulmonary vessels treated?

-P9 L18 The subtraction of bronchial trees from the total air filled volume is described. This leads me to the question whether the value VA provided in the results is the “total” air space volume or just the volume of the terminal airspaces. This is not explained. See point below.

-P9 The variables VL, VA, SA should be introduced and explained in the methods section. What do they consist of? How where they acquired?

-P10 How were layer thickness and diameter determined? The same question goes for the other diameter, thickness, length, etc. values. This should be explained in the methods section. What was the sampling procedure for this? How much deviation was there?

-In the µCT volume renderings, the surfaces of the airspaces are very rough. This is probably due to using a global threshold for segmentation. The result does not represent the smooth surfaces these structures have in real life, as e.g. seen in the provided EM images. This artificial roughness will skew the calculations of surface area (will increase) and volume (will decrease). This should be discussed.

Minor points

-There are inconsistent spellings, e.g. airspaces and air spaces (P2, L9, L11) through the text. In addition, the wording in some sections, esp. in the discussion is rough. Some sentences (e.g. P12, L10-11) do not make sense. Please read the text again carefully and if possible, have a native speaker read through it as well.

-P6 L17f Only the „first weeks of life“ are mentioned, yet later adult animals are studied too. This makes little sense to me.

-P7 L1 The term adult is not defined. At least the age range of the adult animals studied should be given here.

-P10 L5 Please indicate in the text that the value given for the lung volume represents the median of all animals. This goes for the following values too.

-P15 L25f This part is written in the past tense with no apparent reason.

-P17 L16 and L24 Measurements are provided in mm³ and the comparison values in µl. I am aware of the fact that the conversion between two units is 1 to 1; nevertheless, it might improve the readability to give all values in the same unit.

-P30 Table Va is used instead of VA as defined in the main text. The same goes for SA.

-P30 Table The range for 13 dpc seems off. The values in the supplementary table range from 0.36 to 0.66. If I am not mistaken, the range should thus be 0.3 and not 0.12. The only way to reach that value from the numbers provided would be to only calculate the range between the two middle values (0.48 and 0.36). Therefore, this might have been a slight mishap in the calculation process.

Reviewer #2: ---

title: 'Review of "Development of the terminal airspaces in the Gray short-tailed opossum (<monodelphis domestica="">) – 3 D reconstruction by microcomputed tomography" by Kirsten Ferner'

author: David Haberthür

date: 2.11.2023

tags: [review, microct, lung development, opossum]

---

# General remarks

Kirsten Ferner used X-ray computed tomography to investigate the development of the terminal air spaces of the lung in the Gray short-tailed opossum during the first weeks of life.

The study does obtain functional volumes of the air spaces and surface areas of the lung using 3D reconstructions of computed tomography data.

The results show that the development of the lung in Monodelphis domestica resembles both the supposed marsupial and mammalian ancestor.

The study also mentions some inherent flaws of the method, mostly on the sample preparation side, but gives important insights into the development of the opossum lung which closely matches the development of the mammalian lung.

## Major points

Existing literature is - as far as I know - put into context well, and the data also matches the expectation and previous data available.

The main flaws of the study are on the sample preparation side and are well discussed.

Nonetheless, I would like to see more - maybe also graphical - relations to previously published data, to make the acquired data better comparable.

I see some flaws with the presentation of the data, namely that the numbers are presented a median and range in Table 1.

I think that the median is *not* suited when thre are only 2 to 3 values for each timepoint.

I would prefer to see the valued as mean +- standard deviation, and even better in the table.

Also, Fig. 11 plots *all* the values; giving them fully in Table 1 would make it unnecessary to have the supplementary data.

The author does not specify how the data was analyzed and plotted, this should be amended in the methods section.

Figure 3, 5, 8 and 10 show colorful segmentations of the functional lung units, but it is not explained in the text how these units have been extracted from the 3D data.

Is this a built in function of Volume Graphics Studio Max?

I would very much like to read how exactly this was done in section 2.5.

It is not necessary for *this* manuscript, but I would very much like to see a 'development curve' of the volumes of the segmented parts of the lungs, meaning are the volumes of the different lobes or even the different acini developing equally over the course of the studied timeframe, or are there differences evident?

I think this is 'hidden' in the data and could be an interesting follow-up project.

The whole study is providing an overview of the lung development in marsupials and is perfomed scientifically sound, as far as I can judge it.

Once the major points are discussed in more depth and implemented in the manuscript I would recommend to publish the manuscript in PLOS ONE.

## Minor points

Throughout the whole manuscript there are inhomogeneities in the text, which should be corrected.

Three-dimensional data is abbreviated as "3D", "3-D" and "3 D", there should only be one version, I suggest "3D".

Microtomographic imaging is abbreviated as either "Micro-CT" or "µCT", but should only use one version, after it has been introduced for the first time on page 6, line 19.

Mostly, numbers and units are written with "number space unit", but several times without a separating space, this should be homogenized, too.

I think it's customary to use "number space unit".

Number ranges should be given with a en dash, not a hyphen, i.e. not "13-14 days" but "13–14 days".

Sometimes, ranges are given with space and sometimes without.

Approximately is sometimes given as "~ number" and sometimes as "~number", without space.

I would generally write that out instead of stating it with a tilde.

Please homogenize all instances of all these issues throughout the text.

It is stated that the data is "fully available without restriction" but the given link to the data repsitory (https://doi.org/10.7479/cy7h-j182) does *not* work.

## Detailed comments

The detailed comments are given referring to the page and line numbers *printed* in the PDF (e.g. p6, l23), not the real page numbers in the collated PONE PDF.

Most of these detailed comments are suggestions for consideration.

If I think it is necessary to do the change, I write "should be" or "has to be" or otherwise formulate it tersely.

### Abstract

p2, l13: "M.domestica" should be spelled out.

p2, l16 and l18: "... the saccular stage by 7 days" should specify the postnatal days, as well as *Between 28 and 35 days alveolarization started."

p2, l20: I would change "With alveolarization the volume of the air spaces and the surface area for gas exchange increased markedly" to "The volume of the air spaces and the surface area for gas exchange increased markedly with alveolarization."

### Key words

p2, l27: Even though "µCT" is used for microtomographic imaging throughout the manuscript I would use "microCT" as a key word.

### Introduction

p3, l11: I think that "in certain respects" should be "in certain aspects"

p3, l15: Change "born generally" to "generally born"

p3, l20: Can you add a citation for this statement?

p4, l7: "low air-blood" should be "short air-blood"

p4, l12: I think that "studied mainly" would better be written as "mainly studied"

p4, l12: You could also mention mice here in addition to rats, since especially Sonja Mund an Johannes Schittny (which you're citing) have studied the lung develoment in mice.

p5, l26: "remodeling" doesn't seem to be the correct word here.

p6, l4: Replace "underwent" with "undergo", maybe?

p6, l10: Change "...but not the sequence of developmental steps resulting in final lung maturation" to "...but the sequence of developmental steps resulting in final lung maturation are not."

p6, l26: Maybe state that the MfK is in Berlin, *Germany*

p7, l4: "short" should be "shortly"

p7, l6: It is never specified what exact timepoint "adult" refers to. Please state this.

p7, l7: Maybe change "The numbers and specifics of the specimens..." to "All available details of the specimens..."

p7, l9 and l22: You both mention SEM and TEM in the text, but only SEM data is shown. Please clarify.

p7, l17: Plese give as exact times as possible an not just "for a longer period of time"

p7, l18: "In late developmental stages, from 19 dpn to adults the lungs" should be "For late developmental stages, from 19 dpn to adults, the lungs" (with a comma).

p7, l27: No TEM data is shown in the manuscript

p8, l5: Section 2.4 mentions "uCT imaging", here it's only "uCT", please homogenize

p8, l7: I think it is necessary to quickly explain why samples have to be stained. The "general population" does not know why.

p8, l13: "could not be detected" hides a very imporant result of this exploratory preparation step in this manuscript.

I would welcome it very much if this section could be expanded with more information and maybe also a little bit of results.

p8, l16: Replace "could be" with "were"

p8, l21: Tube in "Phoenix nanotom X-ray|s tube" is probably superfluos, and could be replaced with "machine".

p8, l25: As far as I know, all uCT machines worth with "transmission beam".

p8, l26: It would be great if the different scanning parameters could be given in the supplementary materials.

Or are they available in the (non working) data repository?

p9, l3: Who is supplying the Nexus reconstruction software?

p9: It would be nice to have a quick rundown on the sizes of acquired and reconstructed datasets, both in terms of pixels and size on disk in GB

p9, l5: Section title mentions reconstruction, but section deals with segmentation, visualization and data analysis.

p9, l12: "...were colored in different shades of gray...", "appeared black" hides the fact that the tissue density is mapped to gray values.

It's not the coloring of the tissues that is important, but the mapping of the gray value to density.

This has to be explained better.

p9, l17: Change "and could be deducted from" to "and was be deducted from" (or "and was substracted from")

p9, l19: Replace "by the program Volume Graphics" with "with Volume Graphics Studio Max" and specify how this was calculated.

Is this a given function in the software?

p9, l20: "and indicated by mm³ for volume and by mm² for surface area with an accuracy of two digits after the decimal point." is unnecessary and can be discarded.

p9, l21: As said before, median and range don't seem the best way to state the values in the manuscript. Could you consider giving the complete data, or mean and standard deviation?

p10, l9: "are subdivided only a little" seems very unscientific.

Can you specify this a bit better?

p10, l17: Maybe change "the more peripherally located septa" to "the septa located more peripherally"

p10, l20: "of lung development" is unnecessary, maybe completely delete these words?

p10, l22: "appear smoothly walled" is also rather subjective.

Is it possible to measure this, or give concrete visual comparative examples based on the figures?

p10, l24: Maybe change "between 11 and 21 postnatal days" to *between postnatal days 11 and 21"?

p11, l2: Change "vertically standing" to "standing vertically"

p11, l6: I would change "improve" to "increase", as improve is subjective

p11, l6: The "architectural complexity" is not given in Table 1.

Is this corresponding to the 'surface area'?

p11, l9: Change "new" to "newly"

p11, l14: Remove comma after "both".

p12, l13: "Shortly" instead of "short"

p13, l1: Change "indicates the continuous" to "indicates a continuous"

p13, l4: How was the airspace surface calculated exactly?

Depending on the triangulation and voxel size, the influence of the algorithm can have big influences on the outcome of the number.

I think it would be good to discuss this a bit!

p14, l1: "eastern and northern native cat" and others are mentioned in lower case throughout this page, while the Gray marsupial is always mentioned in upper case.

Does this need to be made consistent?

p14, l5: Change "thus" to "those"

p14, l13: "newly" instead of "new"

p17, l4: Add parentheses around "sacculation ... continued alveolarization".

p19, l8 and l9: The values are once given as factor and once given as percent, maybe homogenize.

p19, l15: Does the inflation really cause a higher variation in lung volume or simply a generally lower measured lung volume, as you state above?

p20, l6: No comma after lung

p21, l4: As stated aboce, the given link does *not* work for me

p30: - The table values should be aligned at the decimal points for easier consumption

- I think it would be beneficial to have all *all* values here, and not only in the supplementary materials.

- As said before, choosing to list median and range seems wrong to me for only 2–4 values. I would show mean and standard deviation.

Figure legends in general: You sometimes write "Scale bar = X mm", sometimes "The scale bar X mm", and sometimes that "The Magnification is indicated by the scale bar".

The magnification is not indicated by the scale bar.

Please homogenize all figure legends to stating the scale bar length, or write that the scale bar length is given in the figures, especially when there are scale bars of different length present, as for example in Fig. 4.

I would prefer if all scale bars in the images are labeled (like in Fig. 5 and 6), making it unnecessary to double-check with the legend.

p31, l4: Missing space between strongly and developed

p33, l9: As with 3D, "2 D" should be written homogeneously throughout the manuscript.</monodelphis>

6. PLOS authors have the option to publish the peer review history of their article (what does this mean?). If published, this will include your full peer review and any attached files.

Reviewer #1: No

Reviewer #2: **Yes: **David Haberthür

---

## [Author Response · Author response to Decision Letter 0]

13 Dec 2023

Comments to Reviewer 1:

I thank the referee for his overall positive comments and the critical reading of the manuscript. I tried to address all points raised and revised the paper accordingly. 

Major points

-P7 L8: The SEM pictures were included in the study to show structural details of the 3D architecture of the developing lung, which can be seen with a better resolution and image sharpness in SEM pictures than with the µCT. They were indeed used to make selected measurements of airspace diameter and septum thickness. However, I agree that a proper morphometric measurement should be applied to present reliable data. Therefore, I included morphometric measurements of airspace diameter and septum thickness of the lungs of the scanned µCT-specimens. Using the fractionator method, the µCT-scans of the lungs were digitally sectioned into eight parts, ensuring that the whole lung was sampled. Eight digital pictures were taken from the 2D sections of the lung at the same magnification (ensured by the same scale) for each animal and analysed using Image J software. The program was calibrated with the scale bar and a line was randomly cast over the image of the lung. On each digital photograph 5 airspace diameter and airspace septa intersecting with the line were measured, yielding a total of 40 for each lung. The values for single specimen and group means are presented as mean +- standard deviation in Table 1. A description of the morphometric measurement was included in the method section under 2.6. 

For a better comparison of the SEM to the µCT specimens I added specifics and details (e.g., body weight, fixation, lung section) for the eight SEM specimens in Table 1. The former Table 1 was replaced by the former supplementary table. It now contains all details and results for the single specimens investigated and presents the group means ± standard deviation. 

-P9 L15 For the reconstruction manual tracing on 16-bit pictures (16000 gray scale values) with a tolerance of 1000-1200 gray scale values around the first segmented gray value was used to ensure that the entire airspaces were captured. A paragraph has been added in 2.5.

-P9 L16 The segmentation process and creation of ROIs are explained in more detail under 2.5

The segmentation used a region grower tool, that marks all areas of the same density-value connected to each other to create a region of interest (ROI). The tissue density is mapped to gray values, so that tissues of the same density appear in the same gray scale value. A tolerance of 1000-1200 gray scale values around the first selected gray value of the ROI was given. New marked areas were included in this ROI. With that tool the entire bronchial tree was manually traced, beginning from the extrapulmonary main bronchi to the terminal bronchioles, and it was visually ensured that only air spaces were included. In that way pulmonary blood vessels and other air-filled areas in or between the lung segments were excluded from the segmentation. Calculations of volume and surface area are built in functions of Volume Graphics Studio Max. With segmentation a ROI will be created, which has a certain volume and surface area. The first ROI “bronchial tree” contained the entire bronchial tree of the lung (see Ferner and Mahlow, 2023). The surface area and volume of the ROI “bronchial tree” was calculated by the program. In a next step the ROI of the bronchial tree was copied and then extended to include the terminal air spaces. The resulting ROI “entire air spaces” included all conducting and terminal air spaces of the lung. Volume and surface area were determined for the ROI “entire air spaces”. By subtraction of the surface area and volume of the ROI “bronchial tree” from the ROI “entire air spaces”, the surface area (SA) and volume of the terminal air spaces (VA) resulted. 

-P9 L18 As pointed out above the values VA provided in the results and table 1 are volumes of the terminal air spaces, since the volume of the ROI “bronchial tree” was subtracted from the ROI “entire air spaces”. 

P9 The variables VL, VA and SA have been now introduced and explained in 2.5 of the methods section. I don`t no how exactly Volume Graphics calculates the surface areas and volumes, since it is an inbuilt function of the program, but I believe these are just mathematical calculations of the segmented volumes.

-P10 Morphometric measurements have been conducted to estimate septum thickness and air space diameter. The fractionator method has been applied to ensure that the whole lung was sampled and randomness in the measurement was introduced, as I already have explained in the first point of the reply to the reviewer. A new section (2.6) was included in the methods to explain the morphometry. The mean and standard deviation for the single specimens and age groups as well are given in table 1.

Roughness of µCT volume renderings

Segmentation was carried out with a region grower tool (with a tolerance of 100-1200 gray scale values around the first selected gray value of the ROI). However, since tracing was performed manually it is possible that not all surfaces, especially in the later postnatal stages, might be reproduced perfectly. This leads to an artificial roughness in some segmentations. I agree this might influence the calculations of surface areas and volumes of air spaces. In these cases, the surface area might be overestimated and the volume of terminal airspaces would be lower than in reality. I added a short discussion of this issue in the method section.

Minor points

-“air spaces” are used consistently in the text now. The sentence P12, L10-11 has been deleted. 

- P7 L17 “first weeks of life” are replaces by “postnatal period”

- P7 L1 The term “adult” has been defined now. For the study primi- or multiparous females approximately one year old were used. Since the young Monodelphis (0-14 days) are firmly attached to the maternal teat (removal would be very painful and cause injuries), the mother had to be euthanized to obtain these stages. For efficiency, the mothers were used for lung fixation and dissection.

- P10 L5 A sentence at the beginning of the results section clarifies that all values for volume and surface area are group means of the respective age group.

-P15 L25 The part of the discussion has been revised to be consistent in present tense.

- P17 L16 and L24 The values have been changed from µl to mm³ to provide better comparability.

-P30 Table1 VL, VA and SA have been adjusted to meet the rest of the manuscript.

P30 Table1 The values are now given as means and standard deviation, for single specimens and age group.

Comments to Reviewer 2:

I thank the reviewer for his thoroughly reading of this manuscript and the comments and recommendations. I tried my best to amend the paper according to the reviewer suggestions.

Major points

I followed the reviewer’s suggestion and modified figure 11. It now shows the individual data points for lung and air space volume and surface area of Monodelphis domestica together with morphometric literature data of marsupials and eutherians (data from rat, cattle, pig and sheep a merged to the eutherian data set) to make the acquired data better comparable.

Presentation of the data: I followed the reviewer’s suggestion and modified the presentation of all data to mean +- standard deviation. Table 1 includes now all values of single specimens and group means. The supplementary table is not necessary anymore and will be omitted. 

A sentence to the analysis and plotting of the data in Figure 11 has been added in the method section 2.5 after explaining how VL, VA, SA were obtained.

Colorful segmentations of the functional lung units: A paragraph at the end of 2.5 explains how the colored segmentations of the functional lung units were obtained. The color can be individually chosen for the segmented ROIs (“set interval color”). To set the neighbouring airspaces apart from each other different colors were chosen.

“Development curve of lung lobe volumes”: I agree that this information would be interesting and it might be possibly to generate the information from the entire segmented lungs. For this the ROI of the entire air spaces must be copied and all the parts of the lung deleted that do not belong to the lung lobe of interest. This might be easily done in early postnatal stages since the air spaces are large and well set apart from each other. In later stages the terminal airspaces of the single lung lobes must be confirmed by the course of the lobar and terminal bronchioles. In my opinion for this paper this is going too far. Besides I am not able to do any segmentations at our CT-Lab at the moment, since all computers are shut down after a cyber attack with no time frame given when work will be resumed. But maybe I will follow the reviewer’s suggestion and investigate it as a follow-up project. For the development of the bronchial tree information is available (see Ferner and Mahlow 2023). Not in terms of volumes, but in number of airway branches (Fig. 13) and airway generation (supplementary figure). The results show that the caudal lobes have a higher increase in airway generation and especially in total number of airway branches compared to the other lung lobes.

The given DOI-link for the published data does not work due to continuing restrictions to access the data at the Museum für Naturkunde resulting from the Cyber-attack from mid-October. Since I can not estimate when the link will be working again, I created new Data-Doi’s with figshare (Data: https://doi.org/10.6084/m9.figshare.24764187; Original images: https://doi.org/10.6084/m9.figshare.24763497; 3D-images: https://doi.org/10.6084/m9.figshare.24771213 and videos: https://doi.org/10.6084/m9.figshare.24764397) to make the data publicly available.

Minor points

I followed all reviewer’s suggestion and amended the text accordingly.

-Data availability statement:

I created a Data-DOI (https://doi.org/10.7479/cy7h-j182) containing all original data, images and 3D-animations of the reconstructed terminal airways. Unfortunately, this link is not working at the moment, since the Museum fuer Naturkunde is still suffering from a Cyber-attack that took place mid-october. Since there no official time-line is given when things will run properly again, I decided to create a new Data-DOI (Fig-Share) to provide the data asap.

Detailed Comments:

I followed the reviewer’s suggestion for Abstract, key words and Introduction and amended the text accordingly.

Introduction:

P3, L11, L15: I changed the text accordingly.

P3, L20: The citation Modepalli et al. 2018 has been added.

P4, L7, 12: I changed the text accordingly.

P4, L12: “Mice” and the citation Mund et al. 2008 have been added 

P5, L26: “3D reconstruction” replaced “3D remodeling”.

P6, L4, L10: I changed the text accordingly.

P6, L26: The location of the MfN has been added.

P7, L4: has been done.

P7, L6: The age stage adult has been defined now. Adult animals were generally primi- or multiparous females one year old.

P7, L7: I changed the text accordingly.

P7, L9 and 22: Two sentences clarify why TEM and SEM samples were included in the study. “The TEM samples were used for further ultrastructural analysis, which is not subject to this study”. “The (SEM) samples were viewed and photographed in a scanning electron microscopic (LEO 1450 VP, Carl Zeiss NT GmbH) to see ultrastructural details of the 3D architecture of the lung.”

P7, L17: I specified the fixation times in the text, generally 1-2 days for Bouin and for weeks/months in Karnovsky until µCT scans were performed.

P7, L18: The comma has been inserted.

P7, L27: “The TEM samples were used for further ultrastructural analysis, which is not subject to this study.” This has been stated in the paragraph. See also comments to reviever 1.

P8, L5: "µCT imaging” has replaced “µCT”.

P8, L7: A Paragraph explaining the necessity for staining for µCT-scans has been added at the beginning of 2.3. “Comparative, functional, and developmental studies of animal morphology require accurate visualization of three-dimensional structures, but few widely applicable methods exist for non-destructive whole-volume imaging of animal tissues. µCT imaging in comparative morphology has been used in paleontology, where mineralized tissue, e.g., bones, were scanned. However, µCT-imaging of soft-tissue structures has been limited by the low intrinsic x-ray contrast of non-mineralized tissues. With sufficient contrast imparted to soft tissues, internal soft tissues, such as lung, liver, kidney, heart, intestine, skin and brain, can be made visible with µCT- techniques. With very simple contrast staining µCT imaging produces quantitative, high-resolution, high-contrast volume images of lung tissue. This is possible without destroying the specimens and with possibilities of combining with other preparation and imaging methods (histology or TEM).

Metscher et al. (2009) summarizes several simple and versatile staining methods for µCT-imaging of animal soft tissues, along with advice on tissue fixation and sample preparation. Based on this information, different staining protocols using inorganic iodine and phosphotungstic acid (PTA), were developed, tested and used to produce high-contrast x-ray images of the lung at different age stages (Table 1).”

P8, L13: In my opinion the reviewer’s suggestion to extend this paragraph leads too far. There were no reportable results obtained, the comparison in respect of shrinking resulted in comparison of volumes of PTA or Iodid stained animals (see table 1). We did not perform a scientific study on this subject.

P8, L16: I changed the text accordingly.

P8, L21: It has been changed according the reviewer’s suggestion.

P8, L26: The different kV, µA and projection-settings can be not obtained at the moment, since I do not have access to the VG data due to the shutdown after the cyber-attack at MfN.

P9, L13 Nexus was not used in this study. The sentence has been deleted.

P9, datasets: I can’t give you exact information about the pixel size of the acquired data sets right now, since I cannot open VG files. But the scans are based on a vector image of 1400 x 1480 pixel. The size of the acquired scan-files on disk depends on the size of the scanned sample (between 6 and 15 GB). For example, a small Araldite-block-scan of the whole upper part of a Neonate (e.g., Neonate 1965_1) generates a VG file of 10.6 GB. A whole neonate (2350_7), scanned in liquid results in a VG file of 12.5 GB. An adult lung (2117) resulted in a VG file of 15.1 GB. 

The 3D-images I used for creating the plates Fig. 4,7 and 9 were saved as JPG-files, in a size range of 180-313 KB (Neonate 2350_7) to 203-282KB (Adult 2117). A 3D-rendering of the reconstruction (e.g. neonate video 2350_7; video turning bronchial tree and terminal airspaces) of 30 sec has a size of 52 MB.

P9; L15: The section title reads now: “Segmentation, visualisation and data analysis for 3D reconstruction”

P9, L12: The sentence in question has been deleted. Instead, two sentences explain how the segmentation was performed: “The segmentation used a region grower tool, that marks all areas of the same density-value connected to each other to create a region of interest (ROI). The tissue density is mapped to gray values, so that tissues of the same density appear in the same gray scale value.”

P9, L17: The paragraph has been deleted.

P9, L19 “Volume Graphics Studio Max” has been spelled out. The volume and surface area calculations are a built-in function of VG. With segmentation a ROI will be created, which has a certain volume and surface area. I can’t say, how VG exactly calculates the values. A paragraph has been added in 2.5.

P9, L20: The part of the sentence has been deleted.

P9, L21: I followed the reviewer’s suggestion fully.

P10, L9: The phrase has been deleted.

P10, L17: The phrase has been deleted.

P10, L20: The phrase has been deleted.

P10, L22: The sentence has been rephrased and the subjective statement has been deleted.

P11, L2: I changed the text accordingly.

P11, L6: I changed the text accordingly. The “architectural” complexity has been replaced by “structural” complexity and refers to Fig. 6H.

P11, L9: I changed the text accordingly.

P11, L14: The sentence has been deleted.

P12, L13: I changed the text accordingly.

P13, L1: I changed the text accordingly.

P13, L4: The surface area was calculated by VG Studio Max, I don’t know the exact algorithm of this. 

P14, L11: normally the common species names are written lower case, but for the Gray short-tailed opossum both upper- and lower-case spelling is used. However, I decided to make it consistent and changed the species name to lower case.

P14, L5: I changed the text accordingly.

P14, L13: I changed the text accordingly.

P17, L4: I changed the text accordingly.

P19, L8, L9: I rephrased the sentence to homogenize the cited data.

P19, L15: I agree with the reviewer, that the lung volumes might tend to be lower than in functional lung (as already discussed in the manuscript), but due to differences in inhalation/exhalation status at the time of death variability in lung volume might result in any case. Therefore, I didn’t change the sentence.

P30: I followed the reviewer’s suggestion and table 1 shows now all values and the supplementary table has been omitted. Mean and standard deviation have replaced median and range. 

Figure legends: All scale bars have been labeled in all figures and all statements about scale bars were deleted from the legends, since the scale bars are self-explanatory in the figures.

P31, P33: I followed the reviewer’s suggestions.

---

## [Decision Letter · Decision Letter 1]

8 Jan 2024

PONE-D-23-30643R1Development of the terminal air spaces in the gray short-tailed opossum (Monodelphis domestica) – 3D reconstruction by microcomputed tomographyPLOS ONE

Dear Dr. Ferner,

Thank you for submitting your manuscript to PLOS ONE. After careful consideration, we feel that it has merit but does not fully meet PLOS ONE’s publication criteria as it currently stands. Therefore, we invite you to submit a revised version of the manuscript that addresses the points raised during the review process. The Reviewers have pointed out to some overall minor editing comments that should be corrected first.

We look forward to receiving your revised manuscript.

Kind regards,

Josué Sznitman

Academic Editor

PLOS ONE

Journal Requirements:

Reviewers' comments:

Reviewer's Responses to Questions

**Comments to the Author**

1. If the authors have adequately addressed your comments raised in a previous round of review and you feel that this manuscript is now acceptable for publication, you may indicate that here to bypass the “Comments to the Author” section, enter your conflict of interest statement in the “Confidential to Editor” section, and submit your "Accept" recommendation.

Reviewer #1: (No Response)

Reviewer #2: All comments have been addressed

2. Is the manuscript technically sound, and do the data support the conclusions?

Reviewer #1: Yes

Reviewer #2: Yes

3. Has the statistical analysis been performed appropriately and rigorously? 

Reviewer #1: N/A

Reviewer #2: N/A

4. Have the authors made all data underlying the findings in their manuscript fully available?

Reviewer #1: Yes

Reviewer #2: Yes

5. Is the manuscript presented in an intelligible fashion and written in standard English?

Reviewer #1: Yes

Reviewer #2: Yes

6. Review Comments to the Author

Reviewer #1: The author has addressed the raised issues and reworked the manuscript extensively. Important aspects of the methods are now described in a more detailed way that aids data interpretation and reproducibility. The data presentation has also been improved. This resolves most of my concerns with the first draft of this study.

My remaining issues concern two previously raised main issues where I would still wish for some additional details. Furthermore, there are inconsistencies in the text that should be fixed. Many are minor details that only affect the readability, but also some factual incorrect statements are present. I recommend resubmission after these minor issues are fixed.

I will refer to page and line numbers of the document without tracked changes.

Major issues:

P11 L20f I do not completely understand the segmentation process, as both manual tracing and a region grower tool with a gray scale tolerance are mentioned. I conclude from that, that you did not manually trace the outline of every air space in every 2D image of your stack, but instead traced something like e.g. the center line (please specify) of the airways and extended that marking by region growing to the airway walls. Please elaborate what exactly was manually marked.

P13 L16f The choosing of a random starting point is omitted in the description, but should be mentioned as it is an elementary part of the uniform random sampling scheme employed here. It might be added to the discussion, that the fractionator principle is employed here in a modified form.

P11 L20f 16-bit images contain 2^16 = 65,536 gray scale values, not 16,000

P12 L23f Please add a source for the literature data you refer to here

P25 L1 2.122 cm² is not 26 times 0.028 cm². The factor should be around 76, or one of the surface size values is off.

Minor issues:

P6 L5f X-ray *micro* computed tomography

P6 L21 The abbreviation µCT is introduced here a second time

Table 1 The SD for 13 dpc is missing the “±”

Table 1 SEM specimen data could be moved to its own table (and might be referenced in the text on P9 L1)

Table 1 PTA = *phospho*tungstic acid

P8 L14 “instillation” instead of “installation”?

P9 L25 The abbreviation PTA is introduced here a second time

P10 L19 No need for the “|” here

P10 L18-24 The sentence stretches over seven lines. It might be broken up for increased readability

P10 L21-26 The format *number* *unit* is used here, while it is *number* *space* *unit* everywhere else

P11 L13 One time *number* *unit* is used and right afterwards *number* *space* *unit*

P13 L1f The figure should be referenced here

P13 L10 A fragment of a deleted section is left over here

P13 L13 Inconsistent writing of air space/airspace

P13 L23 “µct” is used instead of “µCT”

P14 L2 “airspace” is used instead of “air space”

P16 L16 The word “staget” might be a typo of “stage”

P16 L21 “14dpn” is used instead of “14 dpn”

P16 L22 “Fig 8K” is used instead of “Fig 8 K”

P16 L24 Use of postnatal days instead of the established abbreviation “dpn”

P17 L10 Use of postnatal days instead of the established abbreviation “dpn”

P17 L12 “8K” the “8” is not needed here, I think

P22 L13 Use of postnatal days instead of the established abbreviation “dpn”

P24 L9 The abbreviation VL has already been introduced in the segmentation section. No need to define it again

P24 L9 For consistency: “2,629.33 mm³” could be used instead of “2629.33 mm³”

P24 L15f The bodyweights for the other marsupial neonates might be added to aid comparison

P24 L23f Introduction of VL, VA and SA is not necessary here, as they have been defined before

P25 L15 28 *dpn* for consistency

P25 L21 Use of postnatal days instead of the established abbreviation “dpn”

P26 L3 Missing space between “35” and “dpn”

Reviewer #2: # General remarks

I thank Kirsten Ferner for improving the manuscript and to reasonably implement the suggestions from both reviewers.

All main points raised by me have been answered, and the manuscript has been improved, inconsistencies have been fixed and additional information has been added.

I would be great if the few minor comments below are corrected in the final version of the manuscript.

Once these are implemented, I would recommend to publish the manuscript in PLOS ONE, and don't have to see the corrected version again.

Table 1: Animals are sometimes listed with a space between the day and 'dpn' and sometimes not. It should be homogenized to being shown *with* a space as it is throughout the manuscript.

Line breaks in the table header should be corrected, but this is probably done in production at PLOS ONE.

p. 8, l. 8: A space is missing between 2 and g.

p. 10, l. 5ff: "The specimens had to be fixed...". I would write that the specimenst "were" fixed in the container with cotton balls to avoid movement artefacts during the scan and completely remove the sharpness part of the sentence. The 'sharpness' of tomographic data is decreased with movement artefacts, but also other factors.

p. 10, l. 18ff: Several instances where a space between number and unit are missing (ms and kV).

p. 11, l. 13: "4dpn" is missing a space.

p. 11, l. 20: [....] carried out on 16-bit images [...] I would remove "(16000 gray scale values)" as 16 bit images can record up to 65536 gray values, not only 16000.

p. 13, l. 17: The mentioned "Fractionator method" could be explained with a citation.

p. 13, l. 25: "(Image J software; National Institutes of Health, USA)." should be cited according to https://imagej.net/contribute/citing and the version used (either Fiji, ImageJ. Also, the software is titled without space between Image and J (also correct in abstract)

p. 16, l. 21: "14dpn" is missing a space.

p. 26, l. 3: "35dpn" is missing a space.

p. 19, l. 1: "...in thickness (12 μm) In the..." is missing a full stop after (12 μm).

Fig. 1-10: Thanks for inlcuding labeled scale bars in all these figures, this greatly helps with readability.

The scalebar of Panel O of Fig. 8 could have a black background behind the text :)

Fig 11: The figure is greatly improved, but rather pixelated, even when downloading the original image from the EditorialManager (https://www.editorialmanager.com/pone/download.aspx?id=34933182&guid=93cb2d79-4577-4e72-9019-cebf7dc85f22&scheme=1).

Its export quality should be increased for publication.

7. PLOS authors have the option to publish the peer review history of their article (what does this mean?). If published, this will include your full peer review and any attached files.

Reviewer #1: No

Reviewer #2: **Yes: **David Haberthür

---

## [Author Response · Author response to Decision Letter 1]

10 Jan 2024

Comments to Reviewer 1:

I thank the referee for his thoroughly reading of the manuscript and the helpful comments. I tried to address all points raised and revised the paper accordingly. 

Major issues:

P11 L20f I do not completely understand the segmentation process, as both manual tracing and a region grower tool with a gray scale tolerance are mentioned. I conclude from that, that you did not manually trace the outline of every air space in every 2D image of your stack, but instead traced something like e.g. the center line (please specify) of the airways and extended that marking by region growing to the airway walls. Please elaborate what exactly was manually marked.

The reviewer is right. I did not draw the lines of the outline of the air spaces manually. Instead, I used the region grower tool to trace the walls of airways and air spaces. Starting from the centerline of the trachea (starting gray value), the region grower tool was extended to the tracheal wall. From there the ROI was extended by scrolling through the image stack and applying region growing to the airway walls and subsequently to the terminal air space walls. I included this sentence in the paragraph on page 12 for clarification.

P13 L16f The choosing of a random starting point is omitted in the description, but should be mentioned as it is an elementary part of the uniform random sampling scheme employed here. It might be added to the discussion, that the fractionator principle is employed here in a modified form.

I followed the reviewer’s suggestion. And amended the text accordingly. A sentence addresses the issues raised: “This requirement is met by choosing a random starting point and employing uniform random sampling using the fractionator principal in a modified form.” 

P11 L20f 16-bit images contain 2^16 = 65,536 gray scale values, not 16,000

I followed the suggestion of reviewer 2 and deleted the gray scale values.

P12 L23f Please add a source for the literature data you refer to here

I added five references.

P25 L1 2.122 cm² is not 26 times 0.028 cm². The factor should be around 76, or one of the surface size values is off. 

I thank the reviewer for the comment. I corrected the factor.

Minor issues:

P6 L5f X-ray *micro* computed tomography

 I replaced “X-ray” by “micro-computed” throughout the manuscript

P6 L21 The abbreviation µCT is introduced here a second time

I deleted this part. 

Table 1 The SD for 13 dpc is missing the “±”

the “±” by 13 dpc has been added.

Table 1 SEM specimen data could be moved to its own table (and might be referenced in the text on P9 L1)

I followed the reviewer’s suggestion and moved the SEM data to an own table (table2).

Table 1 PTA = *phospho*tungstic acid

PTA has been spelled out.

P8 L14 “instillation” instead of “installation”?

The correction has been done.

P9 L25 The abbreviation PTA is introduced here a second time

“phosphor tungstic acid” has been deleted, it is only PTA now.

P10 L19 No need for the “|” here

the “|” has been deleted.

P10 L18-24 The sentence stretches over seven lines. It might be broken up for increased readability

I broke the long sentence into 3 sentences.

P10 L21-26 The format *number* *unit* is used here, while it is *number* *space* *unit* everywhere else

I have added the spaces between number and unit.

P11 L13 One time *number* *unit* is used and right afterwards *number* *space* *unit*

The space has been added.

P13 L1f The figure should be referenced here

I referenced the former figure 11 here following the reviewer’s suggestion. It changed the order of the figures which had to be changed in the manuscript. Figure 11 became Figure 3, and all subsequent figures shifted to a number higher.

P13 L10 A fragment of a deleted section is left over here

The fragment has been deleted.

P13 L13 Inconsistent writing of air space/airspace

“air space” has replaced “airspace”

P13 L23 “µct” is used instead of “µCT”

“µct” has been corrected to “µCT”

P14 L2 “airspace” is used instead of “air space”

“air space” has replaced “airspace”

P16 L16 The word “staget” might be a typo of “stage”

The typo has been corrected.

P16 L21 “14dpn” is used instead of “14 dpn”

The space by 14 dpn has been added.

P16 L22 “Fig 8K” is used instead of “Fig 8 K”

The space has been added in “8 K”

P16 L24 Use of postnatal days instead of the established abbreviation “dpn”

P17 L10 Use of postnatal days instead of the established abbreviation “dpn”

P22 L13 Use of postnatal days instead of the established abbreviation “dpn” P25 L21 Use of postnatal days instead of the established abbreviation “dpn”

 I use “dpn” now for consistency

P17 L12 “8K” the “8” is not needed here, I think

The “8” has been deleted.

P24 L9 The abbreviation VL has already been introduced in the segmentation section. No need to define it again

Lung volume has been deleted, It’s just VL now.

P24 L9 For consistency: “2,629.33 mm³” could be used instead of “2629.33 mm³”

It has been done.

P24 L15f The bodyweights for the other marsupial neonates might be added to aid comparison

I have added body weights for the marsupial neonates.

P24 L23f Introduction of VL, VA and SA is not necessary here, as they have been defined before

(VA), (VL), (SA) has been deleted. 

P25 L15 28 *dpn* for consistency

 I added “dpn” for consistency.

P26 L3 Missing space between “35” and “dpn”

I added the space.

Comments to Reviewer 2:

I thank referee 2 for his in-depth review and the comments. I tried to address all points raised and revised the paper accordingly. 

Table 1: Animals are sometimes listed with a space between the day and 'dpn' and sometimes not. It should be homogenized to being shown *with* a space as it is throughout the manuscript.

Line breaks in the table header should be corrected, but this is probably done in production at PLOS ONE.

The spaces have been inserted.

p. 8, l. 8: A space is missing between 2 and g.

The space has been inserted.

p. 10, l. 5ff: "The specimens had to be fixed...". I would write that the specimenst "were" fixed in the container with cotton balls to avoid movement artefacts during the scan and completely remove the sharpness part of the sentence. The 'sharpness' of tomographic data is decreased with movement artefacts, but also other factors.

I followed the reviewer’s suggestion.

p. 10, l. 18ff: Several instances where a space between number and unit are missing (ms and kV).

The spaces have been inserted.

p. 11, l. 13: "4dpn" is missing a space.

The space has been inserted.

p. 11, l. 20: [....] carried out on 16-bit images [...] I would remove "(16000 gray scale values)" as 16 bit images can record up to 65536 gray values, not only 16000.

I followed the reviewer’s suggestion and deleted the gray scale values.

p. 13, l. 17: The mentioned "Fractionator method" could be explained with a citation.

I inserted a citation for the Fractionator method.

p. 13, l. 25: "(Image J software; National Institutes of Health, USA)." should be cited according to https://imagej.net/contribute/citing and the version used (either Fiji, ImageJ. Also, the software is titled without space between Image and J (also correct in abstract)

I added a citation for the original imageJ version I used for my measurements. All following 

references had to be switched to a higher number than.

p. 16, l. 21: "14dpn" is missing a space.

The space has been inserted.

p. 26, l. 3: "35dpn" is missing a space.

The space has been inserted.

p. 19, l. 1: "...in thickness (12 μm) In the..." is missing a full stop after (12 μm).

I inserted the full stop.

Fig. 1-10: Thanks for inlcuding labeled scale bars in all these figures; this greatly helps with readability.

The scalebar of Panel O of Fig. 8 could have a black background behind the text :)

I gave a black background behind the scale and text.

Fig 11: The figure is greatly improved, but rather pixelated, even when downloading the original image from the EditorialManager. Its export quality should be increased for publication.

I fully agree with the reviewer and have already prepared a figure with higher quality to include in the final submission.

---

## [Editor Report · Decision Letter 2]

15 Jan 2024

Development of the terminal air spaces in the gray short-tailed opossum (Monodelphis domestica) – 3D reconstruction by microcomputed tomography

PONE-D-23-30643R2

Dear Dr. Ferner,

We’re pleased to inform you that your manuscript has been judged scientifically suitable for publication and will be formally accepted for publication once it meets all outstanding technical requirements.

Kind regards,

Josué Sznitman

Academic Editor

PLOS ONE
---

## [Editor Report · Acceptance letter]

8 Feb 2024

PONE-D-23-30643R2 

PLOS ONE

Dear Dr. Ferner, 

I'm pleased to inform you that your manuscript has been deemed suitable for publication in PLOS ONE. Congratulations! Your manuscript is now being handed over to our production team.

Kind regards, 

on behalf of

Prof. Josué Sznitman 

Academic Editor

PLOS ONE